# Mosaic deletion patterns of the human antibody heavy chain gene locus shown by Bayesian haplotyping

Moriah Gidoni[1], Omri Snir[2], Ayelet Peres[1], Pazit Polak[1], Ida Lindeman [2], Ivana Mikocziova[2], Vikas Kumar Sarna[2], Knut E.A. Lundin[2], Christopher Clouser[3], Francois Vigneault[3], Andrew M. Collins[4], Ludvig M. Sollid [2] & Gur Yaari [1]

Analysis of antibody repertoires by high-throughput sequencing is of major importance in understanding adaptive immune responses. Our knowledge of variations in the genomic loci encoding immunoglobulin genes is incomplete, resulting in conflicting *VDJ* gene assignments and biased genotype and haplotype inference. Haplotypes can be inferred using *IGHJ6* heterozygosity, observed in one third of the people. Here, we propose a robust novel method for determining *VDJ* haplotypes by adapting a Bayesian framework. Our method extends haplotype inference to *IGHD*- and *IGHV*-based analysis, enabling inference of deletions and copy number variations in the entire population. To test this method, we generated a multi-individual data set of naive B-cell repertoires, and found allele usage bias, as well as a mosaic, tiled pattern of deleted *IGHD* and *IGHV* genes. The inferred haplotypes may have clinical implications for genetic disease predispositions. Our findings expand the knowledge that can be extracted from antibody repertoire sequencing data.

[1] Faculty of Engineering, Bar Ilan University, 5290002 Ramat Gan, Israel. [2] KG Jebsen Centre for Coeliac Disease Research and Department of Immunology, University of Oslo and Oslo University Hospital, 0372 Oslo, Norway. [3] AbVitro, Inc, Boston 02210 MA, USA. [4] School of Biotechnology and Biomolecular Sciences, University of NSW, Kensington, Sydney, NSW 2052, Australia. Correspondence and requests for materials should be addressed to G.Y. (email: gur.yaari@biu.ac.il)

The success of the immune system in fighting evolving threats depends on its ability to diversify and adapt. In each individual, a repertoire of extremely diverse antigen receptors is carried by T cells and B cells. In B cells, the antigen receptor is a membrane bound immunoglobulin. In effector B cells, i.e., plasma cells, the immunoglobulins are secreted as antibodies to survey the extracellular environment. Antibodies are symmetric molecules with a constant and a variable region. They are built from two identical heavy chains and two identical light chains. The heavy chains are assembled by a complex process involving somatic recombination of a large number of germline-encoded *IGHV*, *IGHD*, and *IGHJ* genes (for simplicity we will refer to them as *V*, *D* and *J* from now onwards), along with junctional diversity that is added at the boundaries where these genes are joined together[1]. Pathogenic antigens are first recognized by lymphocytes carrying these relatively low affinity receptors. Following initial recognition, B cells undergo affinity maturation, which includes cycles of somatic hypermutation and affinity-dependent selection[2]. Thus, the antibody repertoire of an individual stores information about current and past threats that the body has encountered. Studying this diverse repertoire can teach us about fundamental processes underlying the immune system in healthy individuals[3], as well as reveal dysregulation in autoimmune diseases[4–6], infectious diseases[7–9], allergy[10], cancer[11,12], and aging[13].

Dramatic improvements in high-throughput sequencing (HTS) technologies now enable large-scale characterization of adaptive immune receptor repertoires (AIRR-seq)[14,15]. Extracting valuable information from these sequencing data is challenging, and requires tailored computational and statistical tools which are being constantly developed[16]. Much is being invested, especially by the AIRR community[17], in the collection and standardization of data preprocessing and analysis.

Correct assignment of antibody sequences to specific germline *V*, *D*, and *J* genes is a critical step in AIRR-seq analysis. For example, it is the basis for identifying somatic hypermutation, pairing biases, *N* additions and exonuclease removals, determination of gene usage distribution, and studying the link between AIRR-seq data and clinical conditions. Only very few complete or partial sequences of these loci in the human genome have been published thus far[18–22]. The reason for this insufficiency is that these are extremely long (~1.2 Mb) complex regions with many duplications, which impedes usage of traditional methods for sequencing and data interpretations. Because of the difficulty in performing physical sequencing of these loci, several computational tools have been developed for personal genotype inference from AIRR-seq data[3,23–25].

Although germline genotyping by itself is extremely helpful, deeper insight can be gained by going one step further and inferring chromosomal phasing (haplotyping). Since each antibody chain is generated from a single chromosome, it is important to know not only the presence of genes, but also their combination on the chromosomes. For example, inference of haplotype can provide much more accurate information regarding gene deletions and other copy number variations. These appear to be highly common, as shown by Watson et al.[18] by one complete and nine partial haplotype sequencing of the genomic region encoding the antibody heavy chain locus, using BACs and fosmids.

Haplotyping can be computationally inferred from antibody repertoire sequencing data, using a heterozygous *V/D/J* gene as an "anchor" to define the chromosomes. So far, a statistical framework for haplotyping has been developed for *J6*[26,27], which is heterozygous in ~30% of people (alleles *J6*02* and **03*).

Here, we show that reliable haplotyping can also be performed using *D* or *V* genes as anchors (Fig. 1). Haplotype inference is

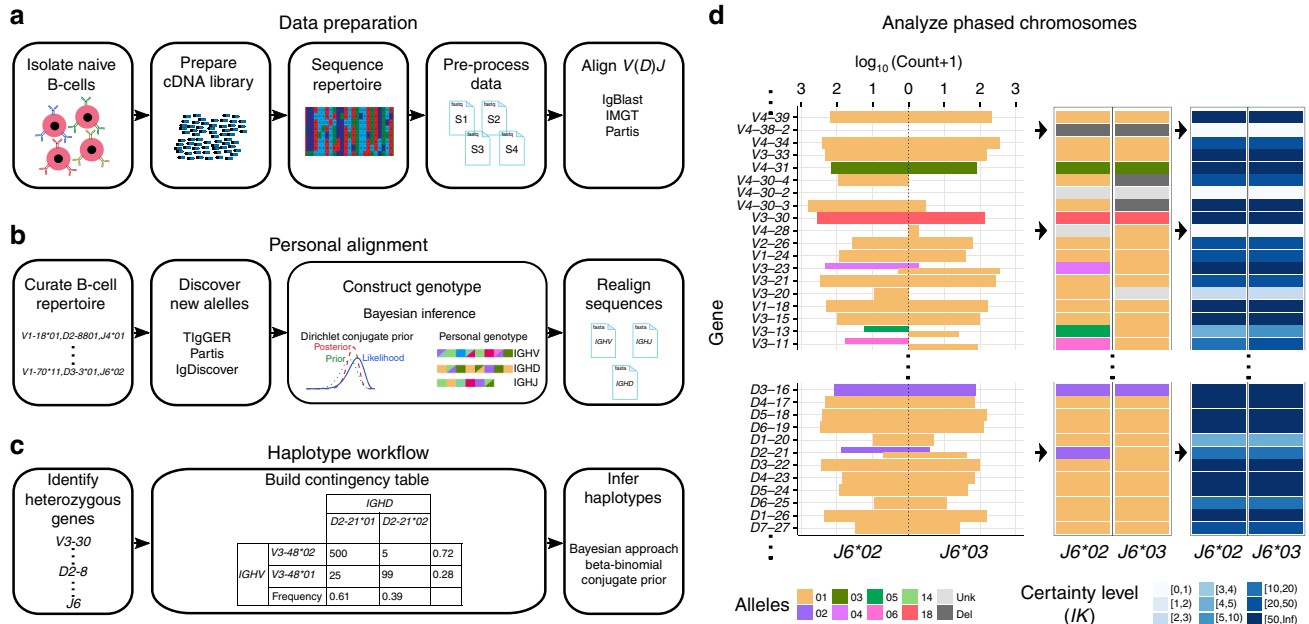

**Fig. 1** Schematic view of the haplotype inference process. **a** Naive B-cells are first isolated, followed by RNA purification. cDNA libraries are prepared, sequenced, the resulting data are pre-processed, and initial *VDJ* alignment is performed. **b** Novel alleles are discovered, and the genotype is constructed. An additional *VDJ* alignment is performed using the constructed genotype. **c** Contingency tables are constructed for *J-V/J-D/V-D* heterozygous gene pairs, and the haplotype is inferred using a Bayesian approach (see Methods). **d** The output can be presented as a phased chromosomes graph. The left panel shows the count of each gene (*Y* axis) that is associated with its paired anchor gene (*X* axis). In this example the anchor gene is *J6*, and the inferred haplotypes are the *V/D* genes. Colors correspond to the different alleles. The thickness of the count bar is inversely proportional to the number of alleles found on the chromosome. The middle panel shows the called *V/D* haplotype, and the right panel shows the certainty level (*IK*) for each haplotype decision. The full haplotype output for the individual in this example is shown in Fig. 7a

performed using a Bayesian approach, and follows an initial deletion identification step based on a binomial test applied to gene usage. Using $D$ or $V$ genes as anchors also enables the $J$ distribution to be examined, and expands the percentage of the population for which it is possible to infer haplotype. We present evidence for allele usage bias, as well as interesting mosaic-like deletion patterns that are common in many individuals and involving multiple genes.

## Results

**Relative gene usage can indicate gene deletions**. Naive B cells from 100 individuals were sorted and their antibody heavy chain variable regions were sequenced using a unique molecular identifier protocol. These data allow us to infer the genetic variability of the antibody heavy chain locus across the largest cohort to date. We exploited the fact that only naive cells were sequenced, to infer and study the characteristics of their germline IGH locus. After filtering out six samples with low coverage (<2000 sequences), personal genotypes of the IGH regions were constructed using a Bayesian genotype approach[28]. To eliminate further potential biases, genotype construction was based on unique sequences with at most three mutations in their $V$ region and no mutations in their $D$ region. Furthermore, only sequences with single assignments for the $V$, $D$, and $J$ genes were used, since sequences with multiple assignments may introduce biases (Supplementary Table 1). In agreement with previous studies[23], genotyping resulted in a five-fold reduction in multiple assignments of a sequence for $V$ genes, and a two-fold reduction for $D$ genes. This reduction was observed by genotyping sequences that were aligned using three different tools: IgBLAST[29], IMGT HighV-QUEST[30], and partis[25] (Supplementary Fig. 1A). ~2% of sequences were initially assigned to genes that were removed during genotyping. They were thus reassigned to genes present in the subject (Supplementary Fig. 1B).

**Deletion patterns of neighboring genes**. Next, we wished to compare the relative usage of different antibody genes across the population. Applying a binomial test (see Methods), we identified deletions in many individuals and multiple genes (Fig. 2a, b). Genes with extremely low expression across all samples were considered indeterminable (NA). In particular, $V1-45$, $V4-28$, and $D6-25$ have very low expression across the vast majority of individuals. It could be that these genes occur only in a very small fraction of individuals with extremely low prevalence. Another possibility is that these are non-functional genes. Looking at the deletions of each sample by itself, several interesting patterns for groups of neighboring genes that are deleted together are observed along the locus (Fig. 2c, d). The most prominent examples are: (i) In 46 of the 47 individuals that lack $V2-70D$, the adjacent gene $V1-69-2$, is also deleted. (ii) In 16 of the 17 individuals that lack $V4-30-2$, the adjacent genes $V4-30-4$ and $V3-30-3$ are also deleted. $V3-30-5$ is located between $V4-30-4$ and $V3-30-3$, we could not infer its deletion, since $V3-30-5$ alleles cannot be differentiated from those of $V3-30$. (iii) Out of 57 individuals that lack $V3-43D$, 56 lack also $V4-38-2$. The sample that lacks only $V3-43D$ had a low relative usage of $V4-38-2$, which was very close to the deletion threshold, but due to a small sample size, could not be inferred as deleted. (iv) Two pairs of genes, $V3-9$ and $V1-8$, and $V5-10-1$ and $V3-64D$, are deleted in a mutually exclusive manner. This pattern has previously been observed for single haplotypes[18,31,32]. Here we show the prevalence of this pattern among a large cohort.

**Ig heavy chain gene heterozygosity landscape**. Inference of a personal genotype allows us to estimate the heterozygosity of genes in the population. We considered genes for which more than one allele is carried by an individual as heterozygous. Up to four distinct alleles in an individual's genotype were allowed, where four alleles would correspond to a mis-named gene duplication with both genes being heterozygous and without sharing between the genes (Fig. 3). It has been previously shown that approximately one third of the population is heterozygous for $J6$[26,27]. Our cohort agrees with this observation with 32/94 heterozygous samples for the 02 and 03 alleles in this gene, and one individual carries alleles 03 and 04, to combine to a total of 33 heterozygous samples. In addition, we identified a large number of heterozygous $V$ genes. Five out of the $V$ genes ($V1-69$, $V3-53$, $V3-48$, $V3-49$, and $V3-11$) were heterozygous in more than 50% of the individuals with a defined genotype, and 19 in more than 20%. Three $D$ genes, $D2-2$, $D2-8$, and $D2-21$ were determined as heterozygous in 2–36% of the population (2, 16, and 28 individuals, respectively, after imposing the 30% threshold as described in the Methods). In the region between $V1-69$ and $V1-46$ (~200 $K$ base pairs) the fraction of heterozygous individuals is dramatically higher than the surrounding regions (Fig. 3a). One possible explanation is that this region is a genomic hotspot for germline evolution giving rise to diverse alleles. Within this region, the three genes, $V3-66$, $V3-64$, and $V4-61$ appear as mostly homozygous, i.e., the same allele is present on both chromosomes. However, there are many single chromosome deletions in $V3-66$, $V3-64$ as shown in the following sections (Supplementary Fig. 2). In the case of $V4-61$, the allele number is less reliable since IMGT may have mis-classified several allele sequences in the $V4-4/V4-59/V4-61$ complex.

We next tested whether in heterozygous individuals, expression of both alleles is similar, or biased towards one of them. For each heterozygous gene, the relative usage of each allele was calculated for each individual (Fig. 4). For the test we considered only samples with a high sequence coverage (>10 $K$), and genes that appeared at a frequency higher than 1%. To statistically address whether there is a biased usage between pairs of alleles that are present in the same individual, a single sample sign test was applied. This test was formulated to consider binary outcomes across the population. For each individual, we asked whether the fraction of the first of the allele pair is larger or smaller than 0.5. Then we noted in how many individuals this fraction is larger than 0.5, and asked how likely this result is to occur by chance. $P$ values were adjusted using the Benjamini-Hochberg method and are referred to as $q$ values. Out of 33 allele pairs (23 genes) that were tested, significant differences were found in 13 allele pairs (11 genes, see Fig. 4). In 10 allele pairs, the preferred allele was significantly more expressed than its partner in all individuals. The range of allelic preferences observed between different individuals is large, most likely due to factors related to their heterogeneous genetic background.

**The single chromosome gene deletion pattern is mosaic like**. To obtain new insights into the $V$ and $D$ gene chromosomal distribution in the population, we inferred the haplotypes of the 33 individuals in our cohort that are heterozygous for $J6$. We applied a Bayesian approach described in the Methods section, and adapted a threshold on the level of confidence to call a deletion ($lK > 3$). Figure 5a shows the distribution of $V$ and $D$ deletions along both chromosomes in these individuals. The deletion likelihood is non-uniform as there are regions along the chromosomes that are more prone to deletions in both chromosomes, and regions that are less prone.

To further investigate the patterns of deletion, we generated a heatmap of $V$ and $D$ deletions (and suspected deletions) for each individual (Fig. 5b). $V1-45$ and $V4-28$ are very rare and therefore their single chromosome deletions are hard to call. The heatmap

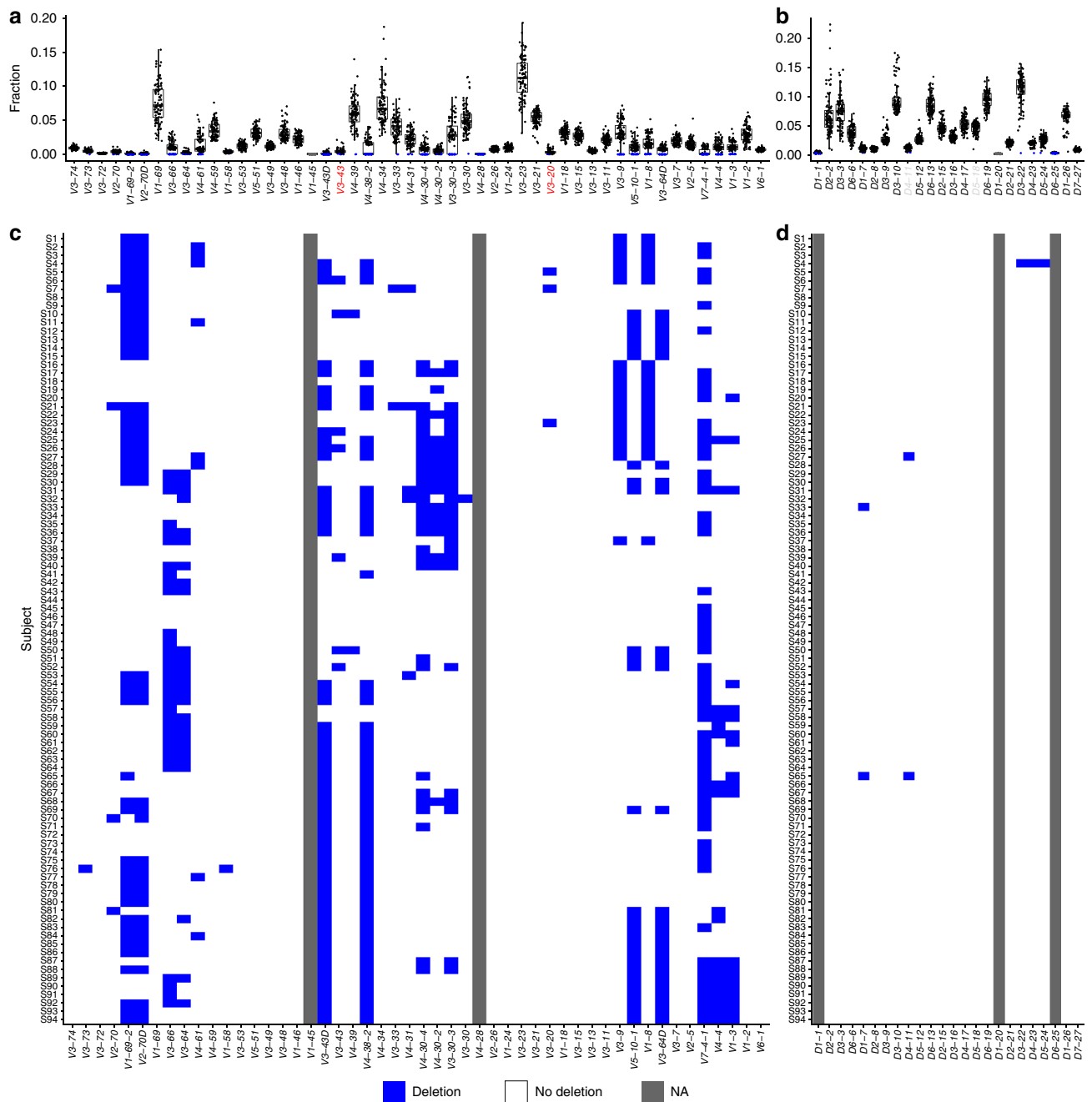

**Fig. 2** Gene deletion inference by relative gene usage. **a**, **b** Box plots of relative gene usage, where each dot represents a single individual. Blue represents deleted genes according to the binomial test (see Methods). The order of genes here and in the following figures is based on their chromosomal location [18]. V gene labels marked in red represent low expressed genes for which deletions are inferred with low certainty. D gene labels marked in light gray represent indistinguishable genes due to high sequence similarity, therefore alignment call is less reliable. **c**, **d** Each row corresponds to an individual. Blue represents a deletion on both chromosomes. Gray represents a gene that was not expressed in more than 90% of individuals (marked as NA). Order of rows was determined based on a hierarchical clustering analysis performed in R using the heatmap function (see Methods). Box plots elements here and onwards are: center line, median; box limits, upper and lower quartiles; whiskers, 1.5x interquartile range; points, all samples

depicts several interesting observations. First, individual S1 has a long deletion stretch in the chromosome carrying J6*02, spanning from V4-28 until V3-64D. This region includes 15 V genes and over 230K base pairs, including the very frequently used V3-23, V3-21, and V3-15. It will be interesting to research any clinical implications this deletion might have on the people carrying it, and if such deletion in a homozygous setting can exist. This pattern might be not so rare, as another individual has been previously shown to have a similar deletion stretch[27].

Second, similar to the pattern observed in both chromosomes (Fig. 2), V3-9 and V1-8 deletion is mutually exclusive with V5-10-1 and V3-64D deletion, in each of the chromosomes. Almost all individuals have one of these pairs deleted in each of the chromosomes. These genes are located sequentially on the DNA. In fact, in 47 of the 94 individuals who passed the sequencing quality filtering criteria, a deletion in both chromosomes of one of these gene pairs was detected using the binomial test (Fig. 2). This is consistent with the assumption that all individuals (not only the

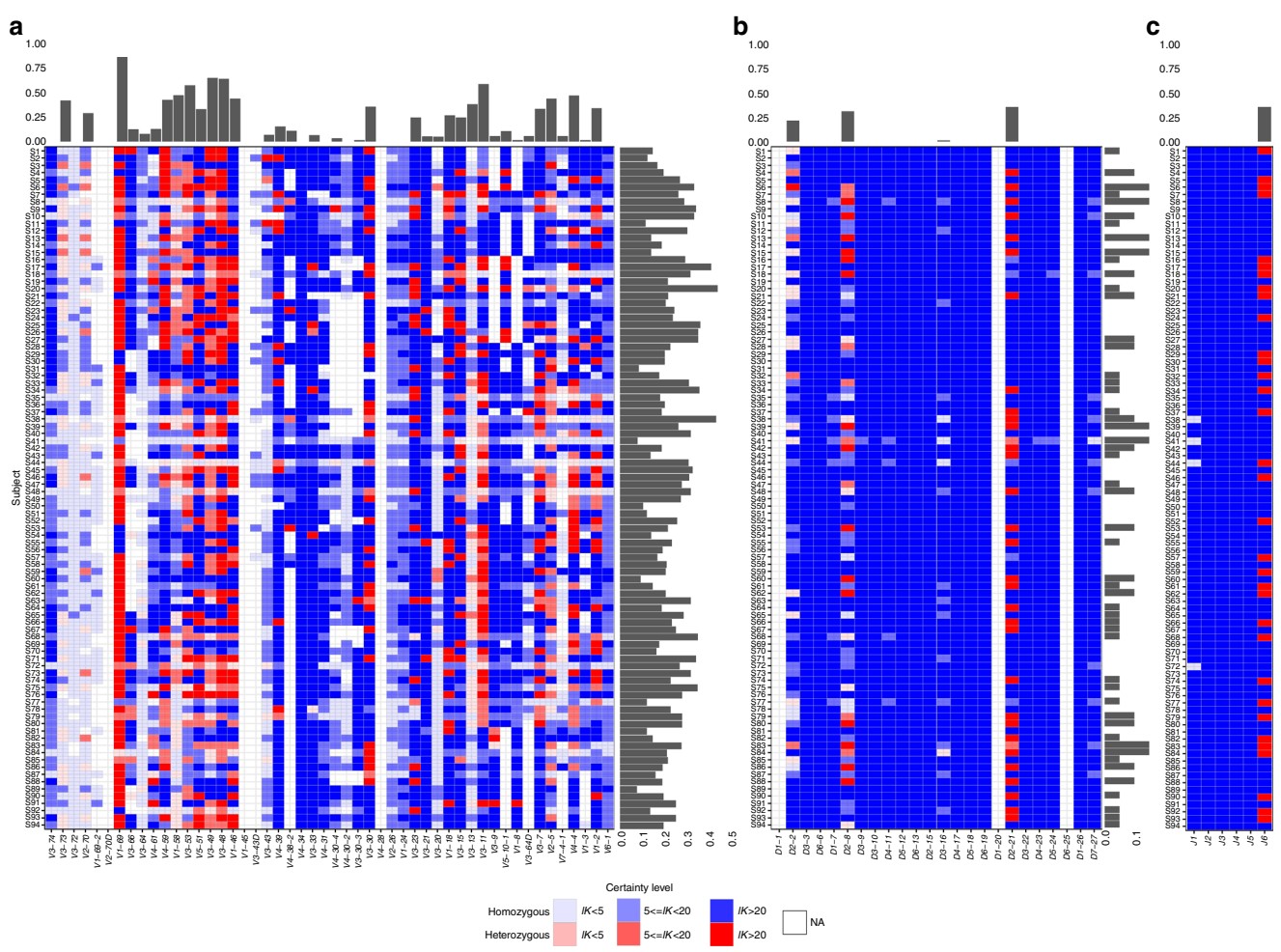

**Fig. 3** Heterozygosity of the IGH genes. Each row represents an individual, and each column represents a V gene (**a**), D gene (**b**), and J gene (**c**). Red shades represent heterozygous genes, and blue shades represent homozygous genes. Transparency corresponds to the certainty level of genotype inference. White represents a gene with too low usage (fewer than 10 sequences, marked as NA) to enable clear genotype inference. Bars on top of each panel represent the ratio between the number of individuals with heterozygous genes and all individuals with a defined genotype for this gene. Bars on the side of each panel represent the fraction of heterozygous genes for each individual out of all genes with a defined genotype

J6 heterozygous ones) have one of these deletions in each chromosome.

Third, nine individuals have deletions in the adjacent genes D3-3 and D6-6. In fact, this deletion stretch might span also D1-7 and D2-8, but we lack the statistical power to say it with confidence. D4-4 and D5-5 have the same sequences as D4-11 and D5-18, respectively, and therefore are not presented here (see Methods). These genes are located within the above deletion stretch. Such a deletion stretch was shown in a previous study[3]. Out of these nine individuals, eight have also a V3-9 and V1-8 deletion, and one individual only has a V5-10-1 and V3-64D deletion (q value of 0.01 by a binomial test). It will be interesting to research the structure of this region in the DNA, and also to find out whether there are any phenotypic differences between these groups.

Fourth, deletions in D3-22 together with D1-26 were observed in the J6*03 chromosome in eight and seven individuals, respectively, and were not observed at all in the J6*02 chromosome.

Since single chromosome deletion inferred by haplotype analysis reflects a deletion polymorphism, we integrated these polymorphisms with other types of heterozygousity to better estimate heterozygosity levels throughout the population (Supplementary Fig. 2).

To verify that the above suspected deletions indeed appear at the genomic level, five selected genes were amplified from genomic DNA (gDNA) of T cells and monocytes using custom-designed gene-specific primers, and PCR products were analyzed by gel electrophoresis. Six individuals from whom gDNA was available were chosen for testing. Primers for amplification and other technical parameters related to the analysis are described in Supplementary Table 2. The target amplicons of V1-8, V3-9, V3-64D, V5-10-1, and V4-38-2 were predicted to be 428, 493, 470, 460, and 500 bp long, respectively. All PCR products were verified by Sanger sequencing and the identity of each sequence was subsequently confirmed by IMGT V-Quest[33]. The findings confirm our inferred observations from the cDNA sequencing data analysis (Supplementary Fig. 3). Individuals S49 and S42 expressed all of the tested genes, and were therefore used as positive controls. Individuals S16 and S4 lack genes V1-8, V3-9 and V4-38-2, but carry V3-64D and V5-10-1. In contrast, individuals S85 and S30 lack genes V3-64D and V5-10-1, but carry V1-8 and V3-9. Moreover, individual S30 also lacks the gene V4-38-2. Two of the individuals, S42 and S49, were found to have one synonymous mutation in V3-64D, which can potentially be a novel allele (Supplementary Table 3).

**Relative gene usage may indicate single chromosome deletions.** Gene deletion identification is of major importance and might

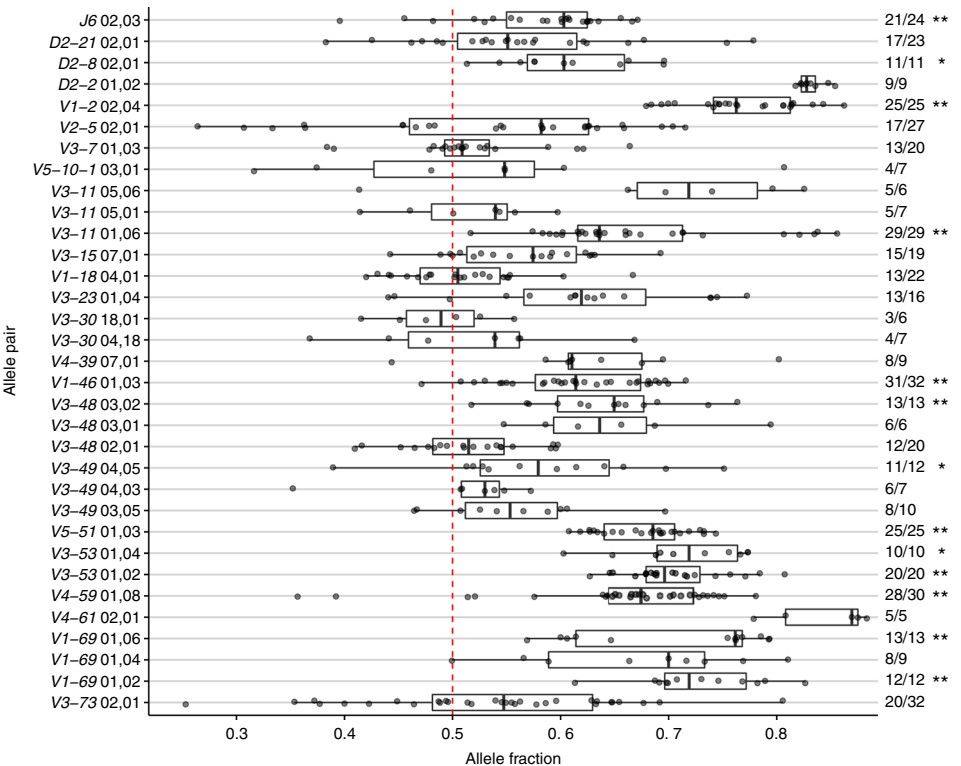

**Fig. 4** Biased allele usage. Box plot of allele pairs relative usage of heterozygous individuals. Only allele pairs which were observed in more than 5 individuals are shown. Each point represents an individual. The allele fraction (X axis) corresponds to the allele that is written first in each row, and is the dominant allele in most individuals. Asterisks indicate allele pairs with a statistically significant difference in the number of individuals with the same dominant allele. Statistical significance was determined with a binomial single sample sign test (see Methods, * indicates p value <0.05, ** indicates p value <0.01)

have critical clinical implications. In the first section of the results, we proposed to use a binomial test to detect deletions from both chromosomes. Haplotype inference offers an additional approach to detect deletions from one of the chromosomes only. We wished to learn the relative gene usage pattern in *J6* heterozygous individuals with single chromosome deletions. Most *V* and *D* genes showed lower usage when one of the genes was identified as deleted from one of the chromosomes according to haplotype inference (Supplementary Fig. 4A, B).

To assess statistically if relative gene usage can indicate gene deletions on a single chromosome, we took the following approach. For each gene, we divided the individuals into two groups, those with no deletions and those with a single chromosome deletion. For each group, we estimated gene usage with a normal distribution. From these estimations, a threshold was derived to call single chromosome deletions (Fig. 6a). By considering different thresholds we created a receiver operating characteristics (ROC) curve for each gene (Fig. 6b), from which we extracted the threshold that yields a false positive rate (FPR) of $\alpha$ (Supplementary Table 4). The obtained sensitivity distributions for $\alpha = 0.01$ and $\alpha = 0.05$ are shown in Fig. 6c. To support our assumption, we estimated the ratios between the mean gene usage of individuals with no gene deletion and individuals with a single chromosome gene deletion (Fig. 6d). The average fold change between the group means is $1.74 \pm 0.18$ (95% confidence interval), which agrees with the hypothesis that genes that are deleted in a single chromosome are expressed at half the frequency. Figure 6e shows gene usage distributions for the two groups for genes that have at least two individuals for each group with a certainty level of $lK >= 10$, along with the derived thresholds. An interesting

exception is *V4-61*, for which the relative usage in individuals with a single chromosome deletion was sometimes higher than in individuals with no deletions. This could be the result of a misclassification of sequences belonging to the *VH4* family. Over 120 sequences in the IMGT reference directory are unmapped. This is true of a majority of *VH4* sequences, which have been given their gene names by sequence alignment, despite the similarities of the *V4-4*, *V4-59* and *V4-61* genes in particular. The problem with this strategy was recently highlighted with the demonstration that the *V4-59*08* sequence maps to both the *V4-59* and *V4-61* loci[34]. This suggests that an individual with an apparent *V4-61* deletion could carry a misnamed sequence at the *V4-61* locus, and it is certainly possible that the high apparent expression of *V4-61* in an individual with a deletion polymorphism is actually high expression of a misnamed sequence.

When *D3-3* is deleted in one chromosome (in our cohort this gene was not deleted from both chromosomes in any individual, see Fig. 2d), it appears to be compensated by higher *D3-10* usage (Fig. 6f, as suggested in ref. [26]). A cutoff of 0.11 on *D3-10* usage correctly classifies all nine individuals with *D3-3* single chromosome deletions. Applying the same cutoff to *J6* homozygous individuals can thus be extrapolated for identifying *D3-3* single chromosome deletions. As shown above, *D3-3* deletion is accompanied by deletions in *D6-6*, *D1-7*, and *D2-8* which are harder to detect due to their low usage. Thus, *D3-10* usage higher than 0.11 implies the above *D* gene deletion stretch.

In the previous section we showed that in *J6* heterozygous individuals, the two *D* genes, *D3-22* and *D1-26* were deleted only in the chromosome carrying *J6*03*. Figure 6g shows the relative usage of these genes for all individuals. All *J6*02* homozygous

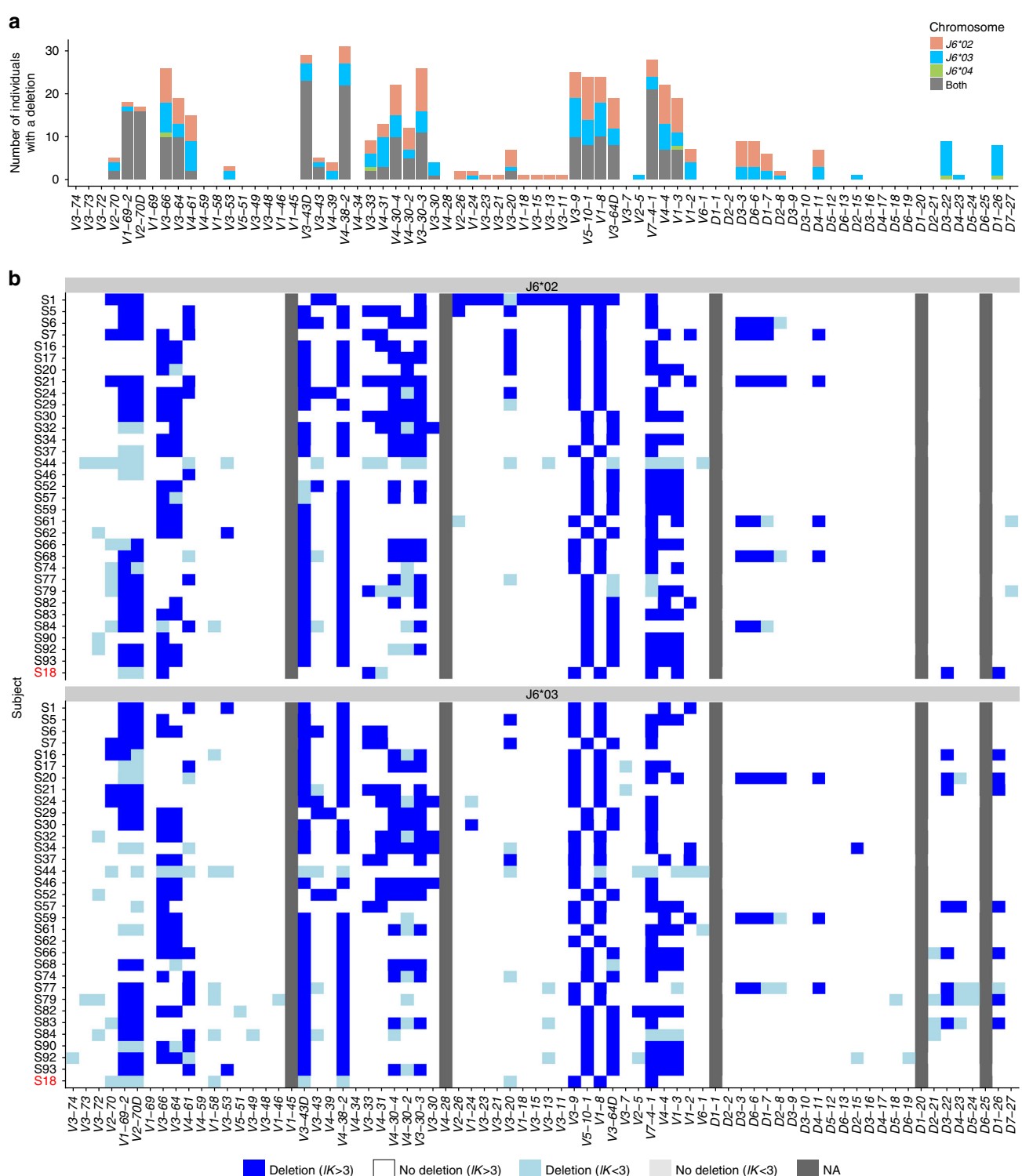

**Fig. 5** Gene deletion inference along each chromosome. **a** The distribution of V and D gene deletions along each chromosome in 33 individuals that are heterozygous for J6, as inferred by haplotype (light red, blue, and green) and by the binomial test (gray). **b** A heatmap of V and D gene deletions and suspected deletions for each of the 33 heterozygous individuals in J6. Each row represents an individual, and each column represents V or D gene. Blue represents a deletion (IK > 3), light blue represents a suspected deletion (IK < 3), and light gray represents no deletion on both chromosomes with low certainty (IK < 3). Dark gray represents a gene with an extremely low usage across all samples. The top panel represents the chromosome on which J6*02 is present, and the bottom panel represents the chromosome on which J6*03 is present. Sample S18, marked in red is heterozygous for J6*03 and J6*04. For this individual, J6*04 was added to the J6*02 panel

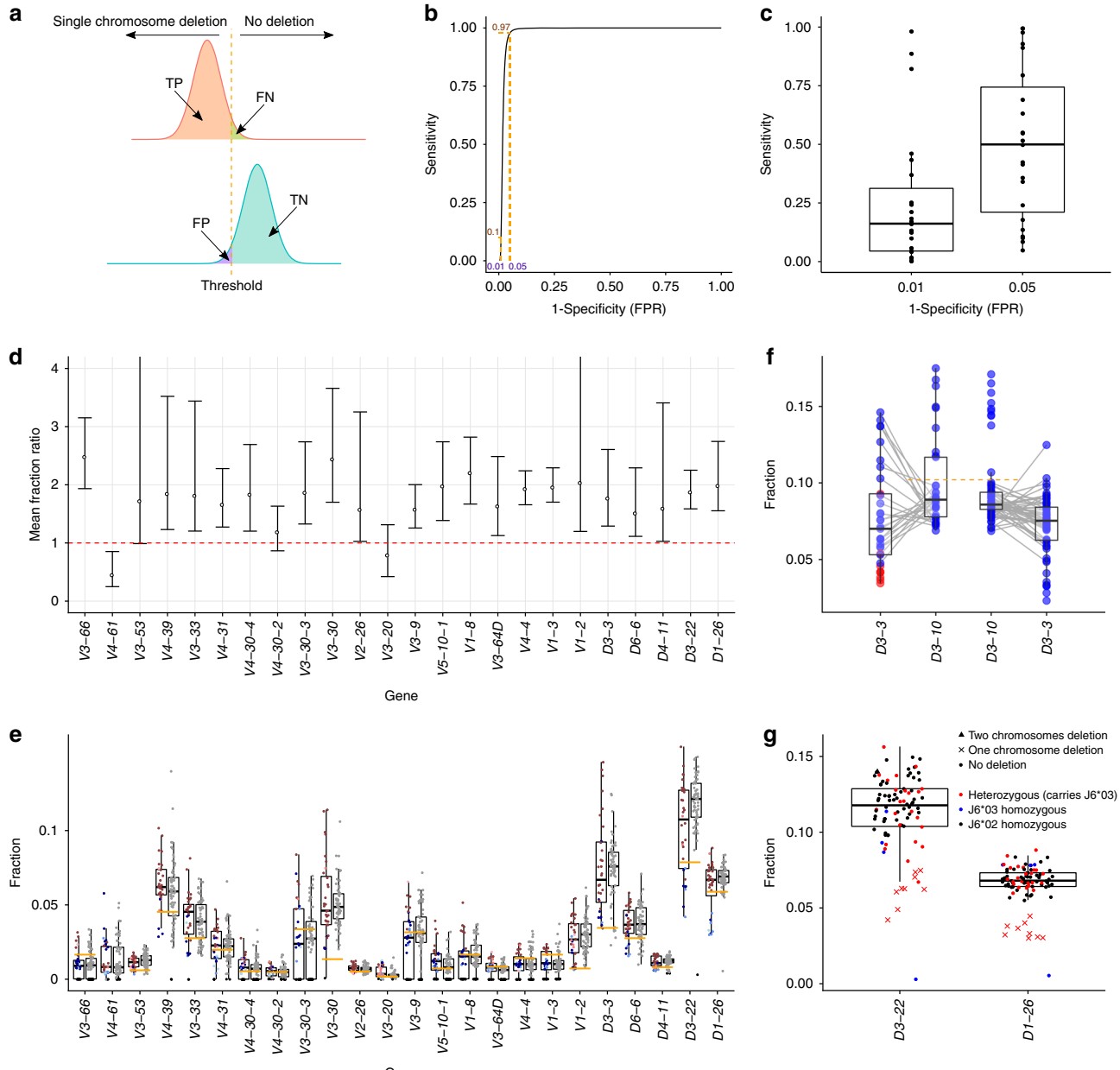

**Fig. 6** Inferring single gene deletions by their relative usage. **a** An example calculation of true positive (TP), true negative (TN), false positive (FP), and false negative (FN) for two t distributions that estimate the gene usage of individuals with zero or one chromosome gene deletion for *V3-66*. **b** An example ROC curve for *V3-66*. **c** Box plots of sensitivity distributions for FPR of $\alpha = 0.01$ (left) and $\alpha = 0.05$ (right). **d** Point estimator and 95% confidence interval for the ratio between the mean gene usage of individuals with no gene deletion and individuals with a single chromosome gene deletion. Genes shown have at least two individuals for each group with a certainty level of $IK >= 10$. **e** Box plots of *V* and *D* gene usage for the genes appearing in **d**. Each gene distribution appears once for the *J6* heterozygous individuals (left) and once for the *J6* homozygous individuals (right). Each dot represents an individual. For the *J6* heterozygous individuals, color corresponds to gene deletion from both chromosomes (black), single chromosome (blue), or no deletion (red). Shades correspond to the certainty level of deletion inference. The orange cutoffs separate individuals with a single chromosome deletion from individuals with no deletions in that gene, with FPR of 0.05. **f** Box plots of *D3-3* and *D3-10* usage for *J6* heterozygous samples (left) and *J6* homozygous samples (right). Gray lines connect between *D3-3* and *D3-10* relative usage of the same individual. Orange cutoffs separate individuals with high and low *D3-10* usage that correspond to single chromosome deletions in *D3-3*. Blue points represent individuals with no *D3-3* deletion, and red points represent individuals with a single chromosome deletion. **g** Box plots of the usage of *D3-22* and *D1-26* for all individuals. Blue and black points represent homozygous individual in *J6*03* and *J6*02* alleles, respectively, red points represent heterozygous individuals carrying *J6*03* allele. The shape of the point represents each individual's gene deletion state

individuals (black) have a higher usage than the usage of the individuals carrying *J6*03* with a single chromosome deletion. In addition, a single individual (S16), with the lowest usage frequency in both, *D3-22* and *D1-26* genes, is *J6*03* homozygous

and has been determined with *D3-22* to *D6-25* gene deletion according to the binomial test. For this sample, *D1-26* usage is just above the binomial test cutoff (0.0056) for being called as deleted, which may imply its deletion if the threshold were

determined for each gene independently. Thus, in this cohort, there were no cases in which D3-22 and D1-26 were deleted from the chromosome carrying J6*02.

**Finding useful D genes for haplotyping.** Compared with V and J assignments, assigning D genes and alleles is challenging and error prone. This is due to the relatively short length of the D genes. As noted above, multiple possible assignments are partially resolved by genotyping, especially for V and J (Supplementary Fig. 1A). The D gene assignment, however, still suffers from a significantly lower credibility. We calculated the allele bias present for the three candidate D genes that can be used for haplotyping (i.e., are heterozygous in a fraction of the population), and observed a distinct set of individuals with highly biased usage (~80%, see Supplementary Fig. 5A, B). Although we saw similar patterns in other genes (Fig. 3), for the purpose of D-based haplotyping we wanted to be conservative, and exclude individuals who present highly biased usage between the two chromosomes based on their D assignments. For this purpose, we built V gene haplotypes for a subset of individuals who are heterozygous for J6 and either D2-2, D2-21, or D2-8. We have plotted the Jaccard distance between the haplotypes of these individuals as a function of allele bias (Supplementary Fig. 5C). Based on this analysis we set up a threshold of 30%, above which the Jaccard distance between the haplotypes is expected to be smaller ($p$ value $<2 \cdot 10^{-4}$ by Wilcoxon test). All of the samples that were initially determined as heterozygous for D2-2 were set as homozygous after applying the 30% cutoff. Haplotype can be inferred only in individuals who carry heterozygous genes, therefore D2-21 and D2-8 emerge here as good candidate anchor genes for haplotyping, due to their relatively high rate of heterozygosity in the population. In our cohort the number of heterozygous individuals increased from 33 to 52 of 94 (Supplementary Fig. 6A). To test the D-based haplotype, we first inferred the haplotype of D by J6 (Supplementary Fig. 6B). Next we inferred the V haplotypes from both anchor genes J6 and D2-21 (Fig. 7a, b respectively). We then calculated the distances between all resulting haplotypes (Jaccard distance <0.1 for the same individual, Supplementary Fig. 5C). These distances were used to generate a hierarchical clustering dendogram (Fig. 7c). The dendogram shows high similarity between the J6-based and D2-21-based haplotypes inferred for the same individual, compared to haplotypes of other individuals. A similar picture is obtained for the D2-8-based haplotypes (Supplementary Fig. 6C), indicating that these D genes can be used for reliable haplotype inference.

**D deletion can be detected using V haplotype inference.** In previous sections we showed how D gene deletions can be inferred either from both chromosomes using a binomial test or from a single chromosome by anchor J6 gene haplotype. As indicated above, J6 heterozygosity prevalence is approximately one third, leaving most of the population without the possibility to infer single chromosome D gene deletions. Since V gene heterozygosity is extremely common (Fig. 3), we pursued the option of inferring a haplotype based on V anchor genes. In our cohort, all individuals are heterozygous in at least two V genes. Thus, using V genes as anchors for haplotype inference could dramatically increase the number of people for which D haplotype can be inferred. However, reliable haplotype inference using V genes as anchors requires a much greater sequencing depth than haplotype inference using J6 gene as an anchor. Since there are far more V genes than J genes, the relative frequencies of the V genes are much lower, making a single anchor V gene haplotype inference more challenging.

To overcome the low number of sequences that connect a given V-D allele pair, we applied an aggregation approach, in which information from several V heterozygous genes was combined to infer D gene deletions. The Bayesian approach utilizing a binomial likelihood and a conjugate beta prior, allows us to use the posterior output of one V-based inference as the prior to the next V-based inference. We do not know in advance on which chromosome each V allele is located. Therefore we attribute all dominant D alleles to the same chromosome. Hence, this $V_{pooled}$ approach is exposed to allele mix, in which contradicting chromosomal alleles assignments are inferred by different V genes.

To assess the power of the $V_{pooled}$ approach, we compared the resulting D gene deletion patterns from $V_{pooled}$ with J6. We compared D genes with minimum mean relative usage of 1.5% in the 32 J6 heterozygous individuals (Fig. 8a left panel, red line). Due to the potential allele mix of the $V_{pooled}$ approach we compared sensitivity and precision for a range of $lK$ cutoffs (Fig. 8b). We identified an $lK$ value ($lK = 12$) which optimized the precision rate (~94% for $lK(J) = 2$ and ~84% for $lK(J) = 7$) with an acceptable price in sensitivity (~46% for $lK(J) = 2$ and ~59% for $lK(J) = 7$). The relatively low levels of sensitivity result from an overall reduction in the number of identified deletions (Fig. 8c). Using the $V_{pooled}$ anchor approach we were able to correctly identify most of the D3-3, D6-6, D3-22, and D1-26 chromosome deletions (Fig. 8d). Applying the same approach to the entire cohort, we identified single chromosome D gene deletions also in J6 homozygous individuals (Supplementary Fig. 7). V anchor gene haplotyping provides an important opportunity to identify D gene chromosome deletions in a much larger proportion of the population than solely by J6. Pooling together several heterozygous V genes as in the suggested $V_{pooled}$ anchor approach, increases the power of D gene deletion identification for moderate sequencing depths.

**Comparison between celiac patients and healthy individuals.** Approximately half of the individuals enrolled in the study have celiac disease (52 out of 100), and these individuals were included to represent genetic variation that might be present among patients with this disease. The study was not powered to perform an association analysis. Yet looking at differences between the two groups, the most prominent difference was single chromosome deletions of the D3-22 and D1-26 genes; D3-22 was deleted in 8 individuals, 7 of whom are celiac disease patients, and D1-26 was deleted in 7 individuals, 6 of whom are celiac disease patients (Supplementary Figs. 8 and 9). These differences are not statistically significant after multiple hypothesis correction.

**Discussion**
Studying the genetic factors that determine the variable regions of B cell and T cell receptors is critical to our understanding of genetic predispositions to diseases. Despite their tremendous importance for the ability of our immune system to fight all sorts of diseases, these regions are understudied and rarely investigated as part of routine disease-association studies. The reason behind this discrimination is technical. The repetitive patterns present in these regions, combined with relatively short reads commonly used in HTS, make it challenging to map them, at both the genotype and the haplotype levels. On the other hand, the technology to produce reliable AIRR-seq data is advancing rapidly, and AIRR-seq studies are gaining popularity. From the early days of AIRR-seq studies, ideas about how to connect these data to genotypes and haplotypes were proposed[3,23–26,35]. Here, we implemented similar ideas in a Bayesian framework that allowed us to: 1. Attribute a certainty level to each result, and 2. Infer

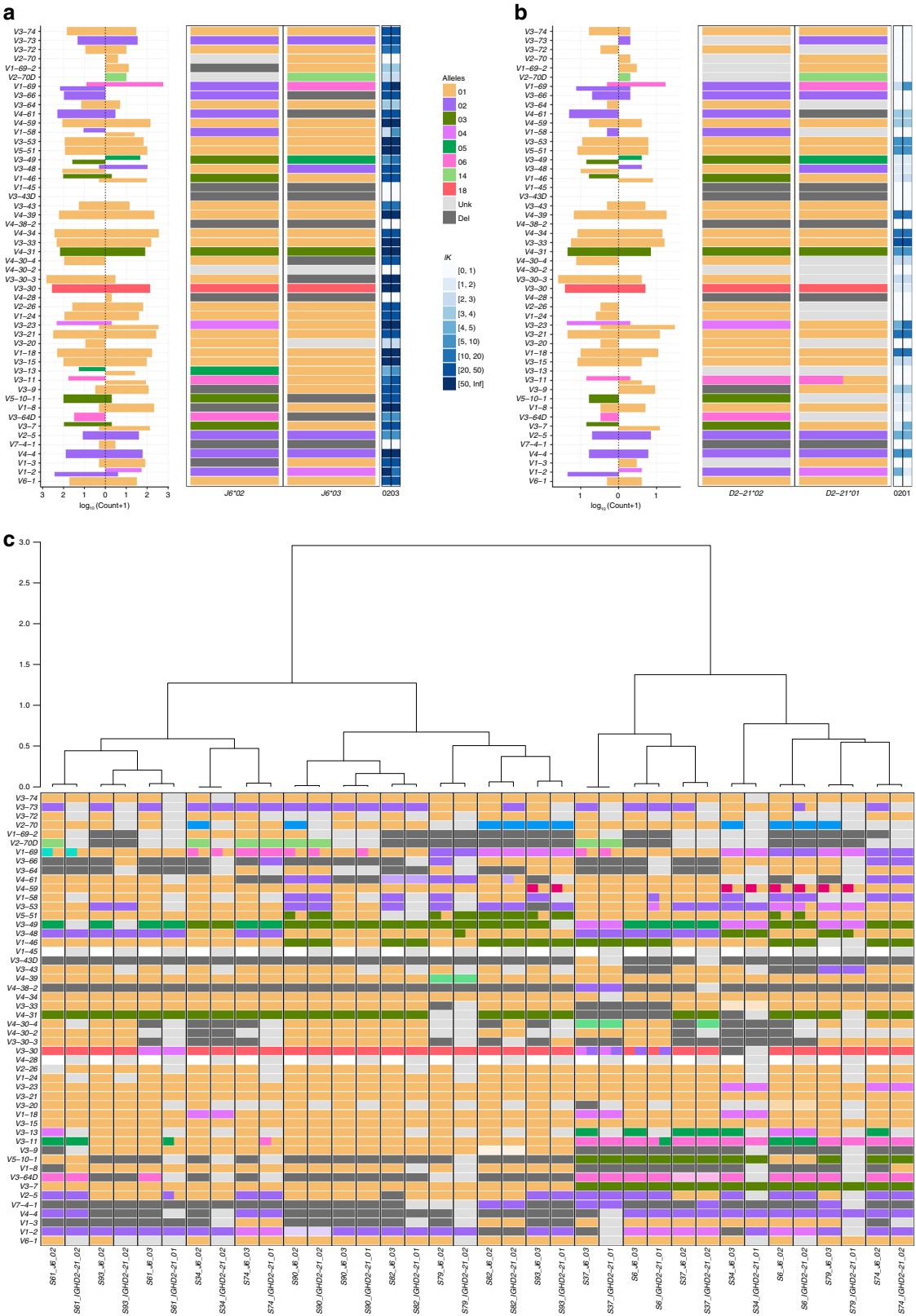

**Fig. 7** *D*-based haplotype inference. **a** Example for *V* haplotype by *J6*. The cutoff between an unknown and a deleted gene on a certain chromosome was determined by *IK* > = 3 (see Methods). **b** Example for *V* haplotype by *D2-21*, for the same individual as in **a**. **c** *V* haplotype map in which each column corresponds to a *J6*-based or *D2-21*-based haplotype of an individual that is heterozygous for both these genes. The order of columns was determined by a hierarchical dendrogram based on the distances between individual haplotypes

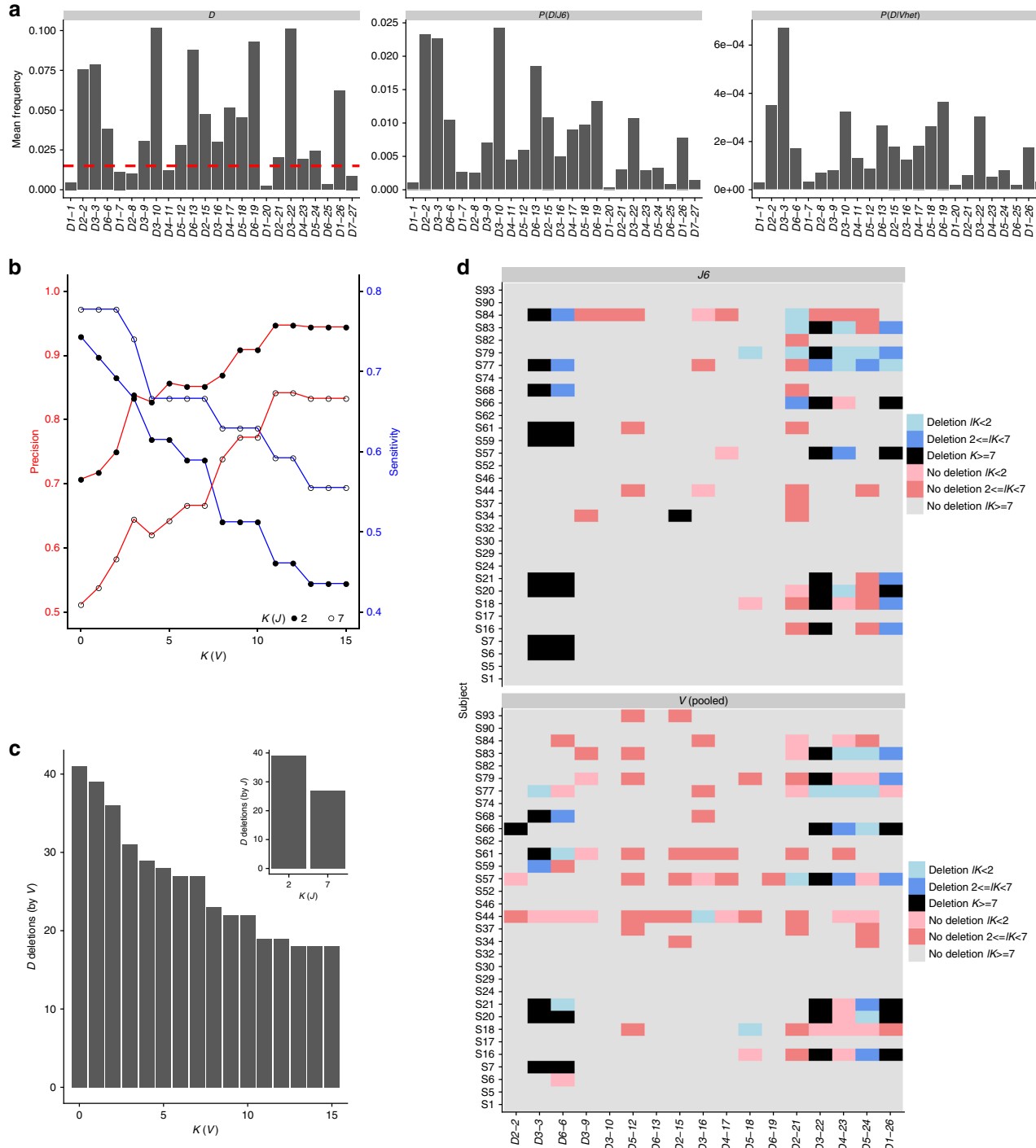

**Fig. 8** Gene deletion inference along each chromosome by multiple *V* genes. A comparison between *D* haplotype inference using a pool of *V* genes vs. *J6* as anchors, in 33 *J6* heterozygous individuals. **a** The mean relative *D* gene usage. Left: mean *D* gene usage. Dashed red line corresponds to the 1.5% threshold which was used to filter out lowly expressed genes for the rest of the analysis presented here. Middle: mean *D* gene usage in sequences containing *J6*. Right: *D* gene usage in sequences containing any heterozygous *V* gene. **b** Precision and sensitivity are described for *D* gene deletions. They are calculated to compare the *D* gene deletions by *J6* as anchor vs. by a pool of *V* genes. Different certainty levels are presented for *V* (*X* axis). Full circles correspond to *lK(J)* >2, and empty circles correspond to *lK(J)* > 7. Precision is shown by the red curves and the left *Y* axis, and sensitivity is shown by the blue curves and the right *Y* axis. **c** The number of *D* gene deletions inferred by pooled *V* (main graph) and by *J6* (subgraph) as a function of the log of the Bayes factor (*lK*). **d** *D* gene deletions inferred by *J6* (upper panel) and by pooled *V* (lower panel). Each row represents an individual, and each column represents a *D* gene. Colors correspond to *lK*s as indicated in the caption. For the presented *V*pooled approach only heterozygous *V* genes with minor allele fraction larger than 30% were included

haplotype based on *V*, *D*, or *J* genes. We generated the largest multiple individual dataset of naive B cells, to date, and applied our method to it. Our study revealed many interesting patterns that are present in the antibody heavy chain locus, and should be investigated further in different populations, various clinical conditions, and using different sequencing technologies.

It had been previously demonstrated that there is a strong bias towards usage of particular genes[36] and between *D* and *J* gene recombinations[37]. In this study we have demonstrated an allele usage bias for various *V*, *D* and *J* genes. Several hypotheses could explain such biases. The first, and most likely one, is differences in the recombination signal sequence (RSS) associated with alleles of the same gene[19,26,38]. Another possibility may be connected to the physical structure of the chromosomes—for example methylation patterns or other epigenetic modifications. Yet another hypothesis is that these biases result from a negative selection process against self-reactive antibodies. It is plausible that certain allele combinations result in self-reacting antibodies, and hence are excluded from the mature B cell repertoire. Note that the latter explanation is not relevant in all cases, since in three allele pairs (*V1-46*03,01*, *V4-59*01,08*, *V5-51*01,03*) the differentiating mutations are silent, i.e., the amino acid sequence is exactly the same.

We showed how gene deletion events on one or both chromosomes can be identified by applying a binomial test to genes with low usage. For the binomial test, we suggested one uniform cutoff for deletion candidates for *V* genes, and another cutoff for *D* genes. This uniform cutoff, however, may not be suitable for all genes and has to be adjusted according to additional parameters. For example, for the *D1-26* gene the cutoff threshold was a bit lower than the usage frequency needed to call it as deleted in individual S24, even though it should have been determined as a deletion by comparing its usage to other individuals (Fig. 6e). For single chromosome deletion detection, the cutoff is even harder to determine. Relying on deletions detected by haplotype, we observed that genes with one chromosome deletion mostly display a lower usage frequency than the same genes in individuals without a deletion. These cutoffs may be influenced by many factors, and as more data become available, factors such as ethnic diversity should be taken into account in setting them. We showed that *V3-9* and *V1-8* deletion is mutually exclusive with *V5-10-1* and *V3-64D* (Fig. 5). This pattern can be utilized also as an anchor for haplotyping.

It is important to note that when we use the "deletion" terminology, we actually mean deletion from the repertoire. This does not necessarily imply that these genes are deleted from the germline DNA. It can be that there were mutations in the non coding region of the allele, the RSS. Such mutations can cause the specific "deleted" alleles not to appear in the repertoire. Hence, to validate inferred deletions and duplication events, sequencing of the genomic region encoding the antibody heavy chain locus is needed. Other major factors that have a strong influence on our approach are the number and type of cells sequenced, and sequencing depth. When sequencing PBMCs for example, a large fraction of the sequenced repertoire will belong to cells that were clonally expanded and have many mutations. This can influence the analysis by creating biases in gene usage estimation due to clonal expansion and allele misassignment due to somatic mutations. Increasing sequencing depth can help by effectively increasing the number of nonmutated cells. Our method was designed to confront the limitations of current sequencing technologies so that it can be used on typical depth datasets. Although small samples might result in undiscovered deletions, the deletions that are discovered in the small samples have a high reliability. The Bayes factor reflects the certainty level in cases of low numbers of

sequences, and different thresholds can be imposed on it to obtain different levels of reliability.

The deletion patterns we discovered include several observations that will be interesting to follow up. First, we observed several genes that are consistently deleted in groups, for example *V1-69-2* and *V2-70D*, as well as *V4-30-4*, *V4-30-2* and *V3-30-3* (see Figs. 2c, 5b). These groups usually consist of adjacent genes, but not always, as in the case of *V3-43D* and *V4-38-2* which are separated by two pseudo genes. Second, there is a deletion in almost all chromosomes of either the gene pair *V3-9* and *V1-8* or *V5-10-1* and *V3-64D*, supporting the hypothesis that these gene pairs are persisting haplotype blocks.

The model parameters used here are based on simplifying assumptions such as a constant probability to miscall an allele ($\varepsilon$). This can be further developed in future studies to be based on empirical data, as outlined recently[39]. In particular, identifying single chromosome gene deletions from relative gene usage, relies on thresholds that we estimated here. These thresholds can be updated as more data accumulates, leading to more accurate estimations.

## Methods

**Library preparation and sequencing**. One hundred individuals from Norway were enrolled in the study; 48 healthy controls (out of which 28 blood bank donors and 20 healthy individuals), and 52 patients with celiac disease. Naive B cells (defined as CD19$^+$, CD27$^-$, IgD$^+$, IgA$^-$ and IgG$^-$) were sorted on a FACSAria flow cytometer (BD) from all 100 individuals (Supplementary Fig. 10). The cells were immediately spun and cell pellets were kept at $-80$ °C until RNA extraction (using RNeasy Midi kit, Qiagen). Participants gave written informed consent. The research is covered by the approval of the Regional Ethical Committee (projects REK 2010/2720 and REK 2011/2472, project leader Knut E. A. Lundin). RNA was reverse-transcribed using an oligo dT primer. An adaptor sequence was added to the 3' end, which contains a universal priming site and a 17-nucleotide unique molecular identifier. Products were purified, followed by PCR using primers targeting the IgD, IgM regions, and the universal adaptor. PCR products were then purified using AMPure XP beads. A second PCR was performed to add the Illumina P5 adaptor to the constant region end, and a sample-indexed P7 adaptor to the universal adaptor. Final products were purified, quantified with a TapeStation (Agilent Genomics), and pooled in equimolar proportions, followed by $2 \times 300$ paired-end sequencing with a 20% PhiX spike on the Illumina MiSeq platform according to the manufacturer's recommendations.

**Verification of selected deletions from genomic DNA**. Amplification of selected genes from gDNA was performed to verify a subset of the predicted deletions. Six patients (S16, S4, S85, S30, S42 and S49) were selected for testing, based on the predicted deletions and availability of gDNA samples (Supplementary Table 2A). Due to high homology between the variable heavy chain genes, gene specificity of primers is often compromised. Therefore, only genes for which specific primers could be designed were selected, namely *V1-8*, *V3-9*, *V3-64D*, *V4-38-2*, and *V5-10-1*. gDNA was isolated from T cells and monocytes (QiaAmp DNA Mini kit, Qiagen) that had been sorted on a FACSAria flow cytometer (BD) from cryo-preserved PBMCs (CD19$^-$, CD3$^+$/CD14$^+$). Gene specific primers were designed using PrimerBLAST[40] (default settings) and reference genomic sequences from IMGT were used as search templates. The primers were ordered from biomers.net GmbH and Eurogentec (Supplementary Table 2B). Target genes were amplified with Q5 Hot Start High-Fidelity DNA Polymerase (NEB) using touch-down PCR with annealing temperature decreasing from 70 °C to 60 °C and 25 cycles in total. PCR products were analyzed by gel electrophoresis and the bands were visualized by staining with Midori Green Advance (Nippon Genetics, Techtum). Then, to validate that the correct sequences were amplified, PCR products were cleaned using the Monarch PCR & DNA Cleanup Kit (5$\mu$g) (NEB), and Sanger sequencing of the products was performed by Eurofins Scientific/GATC. The identity of each sequence was verified by IMGT V-Quest[33].

**Data preprocessing and genotyping**. pRESTO[41] version 0.5.4.0 was applied to produce a high-fidelity repertoire, as previously described[42]. Sequences were then aligned to the *V*, *D*, and *J* genes using IgBLAST[29]. The reference germline was downloaded from IMGT website in December 2017.

Novel alleles were detected by applying TIgGER[23,28] version 0.2.11 to the set of functional sequences. The *V/D/J* gene of a sequence with higher similarity to a novel allele than to the reference gene was reassigned to the novel allele. For each sample a genotype was constructed from sequences with a single assignment (only one best match), using TIgGER adapted for Bayesian approach[28]. Overall, 25 novel *V* alleles were identified and set as part of individuals' genotypes. Next, sequences were realigned according to the inferred personal genotype by IgBLAST, novel

alleles that were part of the genotype, were included in the personal reference. Sequences with more than three mutations in the *V* locus and with at least one mutation in the *D* locus were filtered out leaving on average 86% of the sequences for each sample (range 58–91%). For additional analysis, genotypes were similarly inferred using IMGT/HighV-QUEST[43] version 1.5.7.1 or partis[25] version 0.13.0 (see Supplementary Fig. 1). Five samples with low sequencing depth after filtration (<2000 reads) were discarded from the analysis. Sample names were given after ordering the samples in Fig. 2, using a hierarchical clustering analysis in R with the heatmap function.

**Binomial test for identifying gene deletions.** The *V*, *D* and *J* gene usage varies across genes and individuals. However, in some of the samples, the relative usage of some genes is much lower than in most of the population. To assess if the frequency is low enough to proclaim a certain gene as deleted in an individual, a binomial test was applied. In a given sample, *V* genes with relative frequency below 0.001 were set as candidates for deletion. This threshold was chosen based on the size of the smallest sequence set in the study (~2000). For such a sample, genes that included one or two reads were candidates for deletion by the binomial test. The binomial test has three parameters: number of trials (*N*), number of successes (*x*), and probability of success (*p*). Here, for a given individual, *x* was set to the number of sequences mapped to the *V* gene, *N* to the total number of sequences, and *p* to the lowest relative frequency of this gene among all non-deletion-candidate samples with relative frequencies larger than 0.001. For a given gene, candidate samples with an adjusted *q* value (Benjamini-Hochberg) below 0.01 were marked as deleted. *D* deletion detection was conducted in a similar way, but with a different candidate frequency threshold of 0.005. The larger threshold here was chosen to reflect the reduced reliability of *D* gene assignments. To evaluate the method's robustness for deletion inference in extremely lowly expressed genes, two gamma distributions were estimated for samples with gene usage below and above 0.001 threshold, respectively. In addition, empirical cumulative distribution function curves of gene usage were generated for all samples for a specific gene (Supplementary Fig. 11). Deletions in genes for which bimodal behavior could not be observed were considered less reliable.

**Haplotype inference.** The process is illustrated in Fig. 1. A Bayesian framework based on a binomial likelihood with a conjugate beta prior was adapted to haplotype inference. Using this framework, two biological models were compared. In one model, the considered allele is present on one of the chromosomes only, while in the other model it is present on both chromosomes. For the rest of this paragraph, we assume that we would like to infer the chromosome(s) on which a *V* allele resides, where the chromosomes are identified by the *J* allele they contain. For simplification, we also assume that each sequence represents a unique recombination event, and hence adds one to the number of *V-J* allele pair events. If the considered *V* allele appears with both *J* alleles, inference is expected to tell us that it is present on both chromosomes. If it almost always appears with one of the *J* alleles, we will infer that it is present on one of the chromosomes only. The posterior probability for each *V* allele usage is given by

$$P\left(\vec{\theta}\,|\,\vec{X}\right)_\beta = \frac{P\left(\vec{X}\,|\,\vec{\theta}\right)_{\text{binomial}} \cdot P\left(\vec{\theta}\right)_\beta}{P\left(\vec{X}\right)},$$

where $\vec{\theta}$ is the *V* allele probability distribution, and $\vec{X}$ is a two dimensional vector with components given by this *V* allele's number of occurrences in association with each of the two *J* alleles. Priors were fitted empirically for each individual based on their overall *V* allele usage. The two models are represented by two values of $\vec{\theta}$. For the one chromosome model, we expect all sequences with a given *V* allele to appear together with a specific *J* allele. Hence $\vec{\theta}_1 = \frac{(1+\varepsilon,\varepsilon)}{1+2\varepsilon}$, where $\varepsilon$ accounts for the probability of allele mis-assignment. In the two chromosomes scenario, we expect the *V* allele to appear with both *J* alleles in similar proportions to the *J* allele usage, and hence $\vec{\theta}_2 = \frac{(p+\varepsilon, 1-p+\varepsilon)}{1+2\varepsilon}$, where *p* is the fraction of the dominant *J* allele. The level of confidence in the most probable model is calculated using a Bayes factor, $K = \frac{P(H_{1\text{st}}|\theta)}{P(H_{2\text{nd}}|\theta)}$, where $H_{1\text{st}}$ and $H_{2\text{nd}}$ correspond, to the posteriors of the most and second-most likely models, respectively. The larger the *K*, the greater the certainty in the model. If the evidence is not strong enough, haplotype inference is set to "unknown". Gene deletion events were called on a specific chromosome when for an "unknown" allele the Bayes factor was larger than 1000. For convenience we define $lK = \log_{10}(K)$. All relevant code used in this manuscript is available upon request.

**Determining the heterozygosity cutoff for anchor *D* genes.** To estimate the distance between two haplotypes inferred by different genes, a Jaccard distance was calculated in the following way: (i) For each gene, one minus the ratio between the number of shared alleles over the number of unique alleles from both samples was calculated. For example, for two haplotyped allele assignments *a* and *b* the Jaccard distance was defined as $1 - \frac{a \cap b}{a \cup b}$. Genes that appeared in only one of the samples were excluded. (ii) The overall distance between two haplotypes was calculated as an average of all individual gene distances. Only individuals that had a minimum of 5 genes tested in the Jaccard comparison and each of the genes had a minimum of

five linkages with the question allele were included. Wilcoxon test was used to assert the cutoff that differentiates between the two groups' means. A cutoff of 0.3 was set with a *p*-value < 2e–04.

**Determining thresholds for single chromosome deletions.** Gene usage for haplotyped individuals was used to determine a threshold for a single chromosome deletion. After determining single chromosome deletions in haplotyped individuals for a specific gene, we divided the data for each gene into three groups: 0, 1 or 2 chromosome gene deletions. This division relies upon a minimum certainty level (*lK*), which was set to 10 to be conservative. We then estimated the mean and standard deviation for the groups with zero or one deletion. Since the standard deviation is unknown in advance, the probability distribution from which we can extract confidence intervals is the t distribution with $n-1$ degrees of freedom; i.e., $x \sim \mu + \sigma \cdot t(\nu = n - 1)$ where $\mu$ and $\sigma$ are the estimated mean and standard deviation, respectively. Using these two distributions we calculated TP, TN, FP, and FN for various thresholds (Fig. 6a). From these, we extracted the thresholds for which the FPR is 0.05 and 0.01 (Fig. 6b, c). Confidence intervals for the ratio between gene usage of different groups (Fig. 6d) was calculated with the ttestratio function from the R package mratios[44].

**Comparing *J6*-based and *D2-8*/*D2-21*-based *V* haplotypes.** V haplotypes of samples heterozygous for both *J6* and *D2-8* and for both *J6* and *D2-21* were used to assess the reliability of using *D* genes as anchors for haplotype inference. To compare the similarity between *V* inferred haplotypes, a Jaccard distance was calculated once for samples heterozygous for *J6* and *D2-8* and once for samples heterozygous for *J6* and *D2-21*. The distance between two V haplotypes was calculated in the following way: (i) For each gene, one minus the ratio between the number of shared alleles over the number of unique alleles from both samples was calculated. For example, for two haplotyped allele assignments *a* and *b* the Jaccard distance was defined as $1 - \frac{a \cap b}{a \cup b}$. Genes that appeared in only one of the samples were excluded. (ii) The overall distance between two haplotypes was calculated as an average of all individual gene distances.

**Gene filtration.** For the haplotype inference only functional genes, according to IMGT and NCBI, were used. IMGT ORF and pseudo-genes were removed after genotype inference. *V1-69D* was also removed since for most alleles it is not possible to distinguish it from *V1-69*. *V4-30-1* was removed as well, as IMGT does not have the annotation sequence reference. Two *D* gene pairs have identical sequences: *D4-4*/*D4-11* and *D5-5*/*D5-18*. Therefore only *D4-11* and *D5-18* were used in the inference.

**Single sample sign test.** A special case of the binomial test was used to statistically compare the distribution of values below and above a 0.5 threshold. The *p* values obtained from the test were then corrected using the Benjamini-Hochberg method and referred to as *q* values.

## Data availability
Sequence data has been deposited at the European Nucleotide Archive (ENA), under accession number PRJEB26509 (ERP108501). Additional data that support the findings of this work are available from the corresponding author upon request. A Reporting Summary for this Article is available in the Supplementary section.

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

## Acknowledgements

This research was supported by grants from ISF (grant number 832/16) to G.Y., P.P., A.P. and M.G. and grants from the Research Council of Norway through its Centre of Excellence funding scheme (project number 179573/V40), the South-Eastern Norway Regional Health Authority (project 2016113) and Stiftelsen KG Jebsen (SKGMED-017) to L.M.S.

## Author contributions

G.Y. and L.M.S. conceived and designed the research; V.K.S. and K.E.A.L. collected the samples; O.S., I.L., I.M., C.C. and F.V. carried out the experimental work; M.G., A.P. and G.Y. analyzed the data; M.G., A.P., P.P., A.M.C. and G.Y. wrote the paper. All authors edited the manuscript.

## Additional information

**Competing interests:** The authors declare no competing interests.

