## [Peer Review File · Nature Communications]

Reviewers' comments:

Reviewer #1 (B cell biology, repertoire analysis)(Remarks to the Author):

Gidoni and colleagues describe an approach to infer deletions in the human antibody heavy chain locus starting from high throughput sequencing data of expressed VDJ transcripts. The study extends previous work demonstrating the use of J gene heterozygosity to infer haplotypes to the use of D gene- and V gene-based approaches for haplotype inference. J gene heterozygosity is common in humans (around 30% of humans display J6*02 and J6*03 heterozygosity), but haplotype inference can be applied to more individuals when also exploiting D gene and V gene heterozygosity.

VDJ repertoires from a large number of subjects are sequenced, 100 individuals in total. The authors sort naïve B cells and antibody repertoires are sequenced using the 2 x 300 bp Illumina MiSeq platform. The authors infer frequent deletions of gene segments and apparent non-complementarity between V gene segments in some individuals. The study reveals a mosaic pattern of gene segments in the heavy chain locus and considerable variability between subjects. The results raise interesting questions about how this heterogeneity influences the function of the immune system, a question that is not addressed in the present study.

Comments:

- The study lacks genomic validation of findings from the inference approaches. This is required, at least for a subset of individuals and a subset of the proposed deletions. For example, the authors should seek to confirm the apparent mutual exclusiveness of V3-9, V1-8 and V5-10-1 and V3-64D (Figure 2).
- The study uses a sample set consisting of 48 healthy subjects and 52 subjects diagnosed with celiac disease. The authors state in the beginning of the paper that subjects with celiac disease were included to cover potential genetic variation in this cohort, but this is not further addressed. Can the authors comment on if they observed any differences between the two groups?
- Naïve B cells were sorted prior to sequencing, but details are lacking. The number of naïve B cells sorted from each of the 100 subjects should be included in a Supplementary Table. FACS plots showing representative staining panels should also be shown.
- Figure 1D is incomplete, only about half of the V genes are shown. Why?
- The authors state that they suspect that some genes such as V1-45 and V4-28 are non-functional. Do these genes have open reading frames and intact RSS sequences? An alternative explanation is the V1-45 and V4-28 genes are expressed at very low levels. This should be commented on.
- Figure 2 legend states: "Gray represents a gene with more than 90% deletions". This is inconsistent with the previous statement (see previous point) saying that V1-45 and V4-28 are probably non-functional. The key in Figure 2 indicates that gray means NA and the text on page 3 says gray means N/A. This is confusing, the authors should clarify/correct this.
- On page 5, line 8, the authors claim: "Six out of the V genes (V1-69, V3-53, V3-48, V3-49, V4-28 and V3-11) were heterozygous in more than 50% of the individuals. However, again, earlier they speculated that V4-28 was a non-functional gene. If so, how was heterozygosity determined?"
- In Figure 3A and the corresponding text, the authors state: "In the region between V1-69 and V1-46,

the fraction of heterozygous individuals is dramatically higher than the surrounding regions. This suggests a genomic hot region for germline recombination". This is an ungrounded statement. It may instead reflect that these genes exist in multiple allelic versions as is the case for V1-69, which is marked heterozygous with a high certainty level.

- The authors mention that V4-61 is one of the genes that appears mostly homozygous, but explain later that according to Fig. 6, the 4-61 gene may be mislabelled. If so, it seems imprudent to refer to V4-61 to support an observations.

- There is very little discussion on how the results obtained using inference based on expression from both chromosomes compares to the data obtained from haplotype analysis. What is the sensitivity and specificity of the two methods?

- How reliable is the frequency heterozygosity among the donors if some individuals, including S44, S92 or S95, are assigned the lowest certainty level to all V genes? Supplementary table S1 shows that these 3 individuals stand out in terms of smaller libraries sizes (fewer unique sequences/single VDJ assignments), which makes the resulting data very weak. The authors should discuss the influence on library sizes on the interpretations.

- Also, what is the reason for the small library sizes for some individuals? Donor age, poor cell viability after sorting, poor sequencing or other reasons?

- In Figure 4 and the corresponding text, the authors do not discuss the large range in allelic preference observed between different individuals. What is the basis for this? In several cases this analysis is based on very few sequences assigned to the given alleles,

- On page 9, the authors state: "Most V and D genes showed lower usage when one of the genes was identified as deleted from one of the chromosomes according to haplotype inference." The authors should be careful with "inverse inference" since they base inference of deletions on lower gene usage from the start.

- The binomial test for identifying gene deletions is based on abundance and sequence identity. This difficult to do without a validated genotype for each individual. Details regarding the method for "Identification of subject specific immunoglobulin alleles from expressed repertoire sequencing data" are lacking. The authors refer to a submitted paper (reference 28), which is not sufficient.

- Supplementary Table 1 summarizes the sequencing depth for each individual. On which basis was the "V gene mutation count below 3" criterion chosen?

Minor comments:

- There is a typo in Figure 1A: It is Partis, not Patris.

- The color choices for certainty levels in Fig. 1 and 3 are very busy. Why not use a simple linear color scale instead?

- Throughout the manuscript there are multiple missing words and typos, which need correcting

Reviewer #2 (Bayesian analysis, immunogenetics)(Remarks to the Author):

This paper works to understand the haplotype structure of the antibody heavy chain locus using deep sequencing applied to naive sorted cells from a cohort of 100 individual humans. This is the largest such data set. The authors pair this data set with new haplotype inference algorithms that infer linkage between germline genes, revealing the block-like structure of deletions in this locus.

We very much like this work. Having naive sequence data from almost 100 people is a game-changer for the field, and it's neat to see the new methods other than J anchoring. The analysis of these data is good.

Below we make suggestions about how to polish this gem.

Overall comments

We were surprised that important new methodological advances, i.e. haplotyping using V and D anchoring, happened at the end after all of the rest of the analysis had been done. May we suggest having all of the methods benchmarking happen first, then having an analysis of the resulting haplotype landscape with the combined results of the best tools?

We'd also like to see a more explicit comparison between the methods. For example, if we understand correctly, both the binomial test and the Bayes factor method can be applied to individuals that are heterozygous for J6. Wouldn't it make sense to directly compare the results between the methods for these two techniques? It seems that Figure 5a does some comparison (though note our questions about it below), but it's aggregated into a count that doesn't directly compare the per-subject results. There are also per-subject comparisons (e.g. Figure S3) but this doesn't give us the overall picture. Even if the results don't match perfectly, it'd be interesting to know more. E.g. do the results with huge Bayes factors tend to agree with the binomial-test results more than those with intermediate Bayes factors?

It's also worth thinking about what the overall biological results are from the analysis, one of which is a "mosaic structure." However, the meaning of the word "mosaic" in this context wasn't immediately obvious to these computational biologists. Based on the p.3 text, we *think* that mosaic here refers to having one allele or another, but not both. If you want to have it in the title, please define in the abstract and introduction, and then provide additional discussion about what might lead to these patterns (Are they just persisting haplotype blocks? Is there some reason why there should be exclusion? Can you describe why this is a surprising/interesting result?) From our perspective, though, the coolest part of this paper is that you have done the first large-scale haplotype inference, and an alternate title could focus on that.

The organization of the paper methods is somewhat unusual, with a very short methods section, and a lot of methods appearing in the Results section or in figure legends. We're happy to accept this as a stylistic thing, but there are examples such as Figure S2 where methods important to the paper development are defined in a supplementary figure legend (!). This is not optimal for readability, and having to unpack motivation, methods, and results from this supplementary figure legend took a lot of effort. Having such material in the methods will allow for a more readable flow, starting with the goal

of the analysis and then following with what was done and why.

I (Erick) would love to see the inference part of this paper pushed farther in future work. Right now (Figure 5) you are estimating a Bayes factor for the presence of each V allele on the chromosome for each V gene individually. It would be really nice to get full haplotypes, and the confidence that we have in each one of them. This was the goal in the Elhanati... Walczak paper described below, although they didn't come up with statistical support for their haplotypes. This would end up being a more principled approach to the Vpooled work in the last section. Also, it would be nice to use the fact that the probability of misannotation (epsilon in your setting) depends a lot on what the actual underlying sequences are: a few confident calls for two rather different alleles means a bunch more than a bunch of uncertain calls for similar alleles. I think a fully Bayesian approach to haplotype inference is possible (we outline some ideas in our recent "Bayesian optimist" review paper). Of course, I'm not asking for additional work here, but I hope you will pursue these directions. You could mention such avenues in your Discussion.

There are lots of figures that present genotypes of lists of subjects. Subjects are inherently unordered, which raises the question of how to order individuals in these figures. The sorting of the subjects of course has a large impact on our perception of whether there's patterns in the haplotypes or not. It seems like sometimes the individuals are ordered (Figure 2) and sometimes just ordered by number (E.g. Fig 3, Fig 5).

We feel like some consistency would be good here, and lean towards wanting things sorted to see similarity between genotypes. Here's a way to do this if you want: calculate a pairwise distance matrix between the per-individual results (you are already sensibly using Jaccard in Fig S2), and then do a 1-D multidimensional scaling (there are packages for this in R and Python). Take the induced ordering from this to sort your samples. Then hopefully we can see groups of haplotypes. I note that this is the goal of <https://bitbucket.org/dmarnetto/haplostrips> (<https://besjournals.onlinelibrary.wiley.com/doi/abs/10.1111/2041-210X.12747>), although it doesn't look like that software would actually be convenient for you to use.

Do we know the ethnicities of the donors? Particularly for S65 (different J alleles to everyone else), it would be interesting to know.

I (Erick) wasn't immediately able to find sequence data at the EGA by searching for "ERS2445766". Did I miss something, or will it be released upon submission of the paper? Also, is there a code repository associated with the paper?

Specific comments:

It is so much easier for us and you if you add line numbers to submitted manuscripts.

Abstract:

- suggest "inference to IGHD- and IGHV-based," -> "inference to IGHD- and IGHV-based analysis," or some such
- "We tested our method on the largest..." doesn't actually make it clear that you also generated this data set, which is pretty great!

p.1

- each B and T cell is unique? Hmm... debatable.

p. 2

- suggestion: "Only very few complete or partial"
- suggestion: "So far, a statistical framework"
- suggestion: "This enables looking also at the J distribution"
- suggestion: "We present evidence for"

Relative gene usage[...]

- it wasn't obvious to me staring at table S1 that including multiple assignments introduces biases. Perhaps you can clarify in the table legend or text?
- It would be nice for sequential readers to introduce IK in a sentence or a phrase after you introduce your Bayesian method.
- sentence starting "With the genotype step ~2% of the sequences" as written suggests that 98% of sequences assigned to genes removed during genotyping were not affected? I'm assuming it's a typo, which could be fixed by just removing one stray "that", but suggest something more like: "~2% of sequences were initially assigned to genes that were removed during genotyping. They were thus reassigned to genes present in the subject."
- suggestion: "in many individuals and involving multiple genes"

p. 3 Fig 1

- We like this figure, but feel that it needs to decide if it's an overview figure, or an example results figure. If it's an overview, then 1D should be simplified to make it easily understandable. Reducing the number of germline genes shown would help a lot, which would allow the reader to see cases of heterozygosity more clearly, and give room to add J6*02 and J6*03 labels on the left hand panel of 1D. If you want 1D to show results, then it should be split off to another figure giving a complete example result.

Also, it wasn't obvious at first that the colors of the left-hand panel tell the complete story of the middle two panels (J6*02 and J6*03). If this is going to be an overview figure, then you could have arrows taking us from the count data over to the next two panels to clarify this point. Alternatively, you could drop those next two panels, add the J6*02 and J6*03 labels under the left and right halves of the left panel's x-axis, and add a little symbol showing which low-usage genes were called as deletions (e.g. a grey 'x' or something over the correspond count bar).

- It was difficult to distinguish between the colors-- I'm certain I wouldn't be able to tell the difference between 03 next to a dark bar and 18 next to a light bar. Perhaps try <http://colorbrewer2.org/#type=qualitative&scheme=Set1&n=9> which has 8 colors and a gray?
- Some of the genes on the y-axis have single bars in their row, while others have two bars -- these must be heterozygous? It's confusing to discover this and have to guess, so it'd be nice if it was in the caption. Alternatively, why not just have two rows for genes that're heterozygous?

p. 3

- Sentences after "Specifically:" -- suggest using i,ii,iii,iv rather than A,B,C,D, which also refer to figure panels and I don't think that the points and the panels correspond.
- V1-45, V4-28, D6-25: why is it not equally plausible that these occur only in some individuals and at very low prevalence? Also, it was confusing that in the text these genes were labeled as "non-functional" while in the figure legend they are indicated as being genes with more than 90% deletion. Isn't this an important distinction?
- The take-home message about neighboring genes being deleted together as described in A-D could be more clearly introduced, perhaps with a topic sentence and then a clarification that A-D are examples of this pattern.
- " Further exploration of possible clinical implications" and the previous "Here" sentence sure sound like a Discussion topic rather than a Results one.

p. 5

Ig heavy chain gene heterozygosity landscape

- On the subject of genes with four alleles (i.e. that are presumably misnamed in IMGT) -- many readers would probably find it informative if there was some discussion of how IMGT goes about naming genes and alleles. Especially since such misnaming will have a distinct effect on all (?) of the results in this paper. In other words, all of the results talking about hetero vs homozygosity depend on the extent to which the IMGT distinction between genes and alleles is accurate. The acknowledgment in this paragraph (and at the top of p. 9) that IMGT names are not perfect is important, but it would be good to also discuss how frequent these miscategorizations are, and why you're confident that they're not common enough to materially affect your results.

- To establish a hot region for germline recombination, don't you need to look at more distant correlation between gene use or do some further statistical analysis? It seems that having a region with increased heterozygosity could occur even with a more even mix of ancient haplotypes.

- "Within this region, the three genes, V3-66, V3-64, and V4-61 appear as mostly homozygous. This is not the case, however, because there are many single chromosome deletions in these genes as shown in the following sections." It's not obvious to us that this use of "homozygous" lines up with your definition "We considered genes for which more than one allele is carried by an individual as heterozygous." That is, if a gene is present on one haplotype and deleted on another, isn't it still homozygous by your definition of heterozygous? In any case, more clarity would be appreciated here.

Fig 3

- "fraction of heterozygous genes out of all individuals"-- we feel that it would be clearer to explicitly describe the ratio in terms of a numerator and denominator.

p. 6

- Isn't this just the binomial test? I don't know what a binomial sign test is. The sign test is for paired data. There is a section in the Methods that describes "a special case of the binomial test" but what exactly is this?

- Benjamini-Hochberg: the p-values after applying this procedure should be called q-values. p-values are defined by the fact that they are uniformly distributed under the null. q-values do not satisfy this property, but you are correct that they are the right tool for this job. It might be worth mentioning their interpretation in terms of false discovery rate.

Fig 4

- We suggest dropping B, because A shows the same information in a much more information-rich way. If you want to keep the **, etc, you could just add it after the gene name. In fact, I'm not even sure that the box plots in A add to understanding-- there are an appropriate number of points for the scatter to show everything.

p. 8

Fig 5

A

- We were left scratching our head about this figure. If we understand correctly, this presents the deletion results inferred by haplotype and inferred by binomial test. So then each bin can have the same subject twice? If the intent is to make a comparison between the two methods, it could be done more directly.

caption:

- [typo] caption says both upper and lower panels are for j6*02

p. 9

Relative gene usage may [...]

- While I definitely agree that it's worth checking to see if gene usage can be used to detect single-chromosome deletions, a lot of the detail in this section about particular cases could be removed and summarized by a statement to the effect of "it doesn't really work". A quick glance at Fig 4 shows that allele usage ratios vary by a lot more than a factor of two, confirming the difficulty with such a method. It seems strange to pick the examples where it does seem like there is a separation and put them in a separate figure (6C)-- given this "multiple testing" situation should we believe that the observed separation is meaningful?
- the "it doesn't really work" conclusion seems to be tacitly admitted at the end of p. 10, where you say that the binomial test only works for deletion on both chromosomes (and that j6 anchor has to be used for single-chromosome deletions). If that's the conclusion, then we suggest being more clear about that from the start of this section.

Haplotype can be inferred [...]

- The language in this first paragraph can be tightened up and made more focused-- the current narrative structure is hard to follow. We suggest starting with a topic sentence that describes your goal (finding useful D genes) and then proceeding to results about how well these genes work for haplotype inference. Other details can be moved to the methods section.
- "in a chromosomal linkage map between the alleles of these two genes." We assume you mean D2-21 and D2-8, but please specify.

p. 11

- "Due to the potential allele mix of the V_{pooled} approach" what do you mean by allele mix? That different subjects will have different combinations of V alleles?

p. 13

- First sentence: you mean "genetic factors"
- You are missing a citation of Elhanati, Y., Sethna, Z., Marcou, Q., Callan, C. G., Jr, Mora, T., & Walczak, A. M. (2015). Inferring processes underlying B-cell repertoire diversity. *Philosophical Transactions of the Royal Society of London. Series B, Biological Sciences*, 370(1676). <https://doi.org/10.1098/rstb.2014.0243> who infer haplotypes (you have to see the supplementary material of this paper for details).

p. 14

Library prep:

- golly, it's only on p.14 that we learn that half the cohort has celiacs. Isn't that worth describing earlier? I hope that you will look at association in future work (you could mention that).
- Is "Regy ional Ethical Committee" a typo?

binomial test:

- "the lowest relative frequency of this gene among all non-candidate samples" probably clearer to say "non-deletion-candidate samples" or something, it's hard to remember what they're candidates for

- doesn't this whole procedure depend entirely on the (as far as I can tell) arbitrary choice of candidate frequency threshold? I.e. do the results change if we choose something other than 0.001/0.005 for v/d ?

Data preprocessing:

- not all programs have version numbers
- I'm guessing that you ran partis with the germline inference code turned off, which is now run by default. It'd be worth mentioning.
- It reads a little strangely that first Tigger was applied, then Bayesian Tigger? Ref 28 describes some additional improvements to the core Tigger algorithm, and I'm guessing that this new code was run from the start.

Binomial test:

- suggest "usage of different genes" -> "usage of some genes"

Haplotype inference

- "Each sequence represents a unique recombination event"-- this is an assumption, especially given that we're just looking at heavy chains (IIUC there is proliferation of B cells just after heavy chain rearrangement to get multiple chances to rearrange light chains).
- "by the J allele that is present on them" suggestion: "by the J allele they contain"
- "X is a two dimensional vector with the number of sequences that this V allele appeared with ..."
suggest: "X is a two-dimensional vector with components given by this V allele's number of occurrences with each of the two J alleles"
- how was a Bayes factor of 1000 chosen as the cutoff? It's a bunch stronger than the Kass-Raftery "decisive" cutoff from their foundational paper.

Bayesian Tigger paper:

We understand that we aren't meant to review this paper, but thought we would just add a few minor comments, which hopefully will be helpful.

- the formula for the posterior distribution is missing a P
- The posterior conditions on the data, but here it's written using v , which is an "or". Don't you want a vertical bar?

Review co-authored by Duncan Ralph and Erick Matsen. Feel free to contact us regarding the review.

Reviewer #1:

Gidoni and colleagues describe an approach to infer deletions in the human antibody heavy chain locus starting from high throughput sequencing data of expressed VDJ transcripts. The study extends previous work demonstrating the use of J gene heterozygosity to infer haplotypes to the use of D gene- and V gene-based approaches for haplotype inference. J gene heterozygosity is common in humans (around 30% of humans display J6*02 and J6*03 heterozygosity), but haplotype inference can be applied to more individuals when also exploiting D gene and V gene heterozygosity.

VDJ repertoires from a large number of subjects are sequenced, 100 individuals in total. The authors sort naïve B cells and antibody repertoires are sequenced using the 2 x 300 bp Illumina MiSeq platform. The authors infer frequent deletions of gene segments and apparent non-complementarity between V gene segments in some individuals. The study reveals a mosaic pattern of gene segments in the heavy chain locus and considerable variability between subjects. The results raise interesting questions about how this heterogeneity influences the function of the immune system, a question that is not addressed in the present study.

Thank you for the careful and thorough review of our work. We feel that the manuscript greatly improved as a consequence.

Comments:

- The study lacks genomic validation of findings from the inference approaches. This is required, at least for a subset of individuals and a subset of the proposed deletions. For example, the authors should seek to confirm the apparent mutual exclusiveness of V3-9, V1-8 and V5-10-1 and V3-64D (Figure 2).
 - This is an excellent point, that we have pursued in this revision. We have validated the inferred deletion patterns by sequencing genomic DNA of five genes from six individuals that have different inferred deletion patterns for these genes. New text describing this validation was added to the results and methods sections, and a new figure (S5) shows the validation results.
- The study uses a sample set consisting of 48 healthy subjects and 52 subjects diagnosed with celiac disease. The authors state in the beginning of the paper that subjects with celiac disease were included to cover potential genetic variation in this cohort, but this is not further addressed. Can the authors comment on if they observed any differences between the two groups?
 - The subjects with celiac disease were included to represent genetic variation that might be present among patients with this disease, and we did not power our study to perform a proper association analysis. In the revised manuscript we have nevertheless looked at differences between patients and controls, and we present the results in the new figures S7 and S8. The most prominent difference was single chromosome deletions of the D3-22 and D1-26 genes; D3-22 was deleted in 8 subjects, 7 of whom are celiac disease patients, and D1-26 was deleted in 7 subjects, 6 of whom are celiac disease patients. The differences are not statistically significant after multiple hypothesis correction.
- Naive B cells were sorted prior to sequencing, but details are lacking. The number of naïve B cells sorted from each of the 100 subjects should be included in a Supplementary Table. FACS plots showing representative staining panels should also be shown.

- The required information was added to table S1 and a new supplementary figure (S9) was added that includes a representative FACS plot and the gating strategy.

- Figure 1D is incomplete, only about half of the V genes are shown. Why?

- This is an illustration figure, and we cropped the original graph for aesthetic reasons. A full graph is shown in figure S5c. To avoid confusion we added a reference in the legends of Figure 1D to figure S5c.

- The authors state that they suspect that some genes such as V1-45 and V4-28 are non-functional. Do these genes have open reading frames and intact RSS sequences? An alternative explanation is the V1-45 and V4-28 genes are expressed at very low levels. This should be commented on.

- We agree with this comment. We do not know whether or not these genes are functional or not, and therefore we rephrased the relevant section in the results.

- Figure 2 legend states: “Gray represents a gene with more than 90% deletions”. This is inconsistent with the previous statement (see previous point) saying that V1-45 and V4-28 are probably non-functional. The key in Figure 2 indicates that gray means NA and the text on page 3 says gray means N/A. This is confusing, the authors should clarify/correct this.

- This line has been changed to avoid confusions. We also changed the text so now it says "NA" everywhere and not "N/A".

- On page 5, line 8, the authors claim: “Six out of the V genes (V1-69, V3-53, V3-48, V3-49, V4-28 and V3-11) were heterozygous in more than 50% of the individuals. However, again, earlier they speculated that V4-28 was a non-functional gene. If so, how was heterozygosity determined?

- We thank the reviewer for pointing out this mistake. We deleted V4-28 from the list.

- In Figure 3A and the corresponding text, the authors state: “In the region between V1-69 and V1-46, the fraction of heterozygous individuals is dramatically higher than the surrounding regions. This suggests a genomic hot region for germline recombination”. This is an ungrounded statement. It may instead reflect that these genes exist in multiple allelic

versions as is the case for V1-69, which is marked heterozygous with a high certainty level.

- Indeed the original statement was too conclusive, and thus we moderated it. Regarding the interesting hypothesis raised here, we plotted the number of known alleles from IMGT for each gene, and did not find similar pattern. See figure below:

- The authors mention that V4-61 is one of the genes that appears mostly homozygous, but explain later that according to Fig. 6, the 4-61 gene may be mislabelled. If so, it seems imprudent to refer to V4-61 to support an observations.

- This is correct. We modified the text to reflect this point.

- There is very little discussion on how the results obtained using inference based on expression from both chromosomes compares to the data obtained from haplotype analysis. What is the sensitivity and specificity of the two methods?

- The binomial deletion inference and the Bayesian haplotype inference methods are not comparable. The binomial method is applied before haplotype inference, and is able to detect genes that are deleted from both chromosomes. Once the deleted genes are detected, the next step is haplotype inference of the non-deleted genes. This is done by applying the Bayesian approach with an anchor gene (J6, D2-21, D2-8 or Vpooled). The method's certainty level is estimated by a Bayes factor. By combining both methods, a more accurate and full picture of the gene usage, allelic usage, and haplotype can be acquired. To avoid confusion we added text to the introduction that clarifies the relationship between the two methods.

- How reliable is the frequency heterozygosity among the donors if some individuals, including S44, S92 or S95, are assigned the lowest certainty level to all V genes? Supplementary table S1 shows that these 3 individuals stand out in terms of smaller libraries sizes (fewer unique sequences/single VDJ assignments), which makes the resulting data very weak. The authors should discuss the influence on library sizes on the interpretations.

- We agree with this observation, and edited the discussion to include the following sentence: "Other major factors that have a strong influence on our approach are the number and type of cells sequenced, and sequencing depth."

- Also, what is the reason for the small library sizes for some individuals? Donor age, poor cell viability after sorting, poor sequencing or other reasons?

- We have looked deeply into the data we have about the subjects, including age, RIN score, RNA concentration, disease status and other parameters. We could not find any exceptional parameter, and therefore we assume that the small library size was due to low cDNA or PCR efficiency. RIN score and number of sorted cells were added to table S1.

- In Figure 4 and the corresponding text, the authors do not discuss the large range in allelic preference observed between different individuals. What is the basis for this? In several cases this analysis is based on very few sequences assigned to the given alleles.

- We agree with this observation and added it to the text.

- On page 9, the authors state: “Most V and D genes showed lower usage when one of the genes was identified as deleted from one of the chromosomes according to haplotype inference.” The authors should be careful with "inverse inference" since they base inference of deletions on lower gene usage from the start.

- Thank you for pointing this out. The inference of single chromosome deletions is not based only on usage, but mainly on the relative frequency of pairing with the anchor gene. Figure 6a shows that single chromosome deletions are found even in genes with high relative usage. For example V4-61, V3-33, V3-30-3, V2-5. Also, there are cases in which a gene has a higher relative usage compared with other genes in the same individual, and still be classified as a single chromosome deletion, since the expressed alleles are paired only to one of the anchor alleles.

- The binomial test for identifying gene deletions is based on abundance and sequence identity. This difficult to do without a validated genotype for each individual. Details regarding the method for “Identification of subject specific immunoglobulin alleles from expressed repertoire sequencing data” are lacking. The authors refer to a submitted paper (reference 28), which is not sufficient.

- We added the cited manuscript to the submission. We will deposit this manuscript in the bioarxiv by September 1st.

- Supplementary Table 1 summarizes the sequencing depth for each individual. On which basis was the “V gene mutation count below 3” criterion chosen?

- Our data set is sorted naïve B cells. We determined the cutoff in order to remove any non-naïve B cells that may have been wrongfully included in our data set due to sorting errors. We set the cutoff to 3 mutations to maintain a reliable naïve repertoire, while not excluding antibodies that may have gained reverse transcription errors (amplification or sequencing errors should be corrected using UMIs). The choice of 3 as a cutoff was made to include ~90% of the sequences. See the following figure:

Minor comments:

- There is a typo in Figure 1A: It is Partis, not Patris.

- Corrected.

- The color choices for certainty levels in Fig. 1 and 3 are very busy. Why not use a simple linear color scale instead?

- Certainty levels determined the transparency levels in these figures in a linear fashion. We are open to suggestion regarding other color schemes.

- Throughout the manuscript there are multiple missing words and typos, which need correcting.

- We went over the manuscript again and corrected all typos that we saw.

Reviewer #2:

This paper works to understand the haplotype structure of the antibody heavy chain locus using deep sequencing applied to naive sorted cells from a cohort of 100 individual humans. This is the largest such data set. The authors pair this data set with new haplotype inference algorithms that infer linkage between germline genes, revealing the block-like structure of deletions in this locus.

We very much like this work. Having naive sequence data from almost 100 people is a game-changer for the field, and it's neat to see the new methods other than J anchoring. The analysis of these data is good.

Below we make suggestions about how to polish this gem.

Thank you very much for the positive evaluation of our manuscript, and all the constructive comments. We feel that the manuscript greatly improved as a consequence.

Overall comments

- We were surprised that important new methodological advances, i.e. haplotyping using V and D anchoring, happened at the end after all of the rest of the analysis had been done. May we suggest having all of the methods benchmarking happen first, then having an analysis of the resulting haplotype landscape with the combined results of the best tools?
 - We were debating a lot about the order of presentation in this manuscript. As detailed in the reply to the next comment, haplotyping comes after the binomial test, and we did not want to “lose” the readers by starting with more technical sections describing the methods.
- We'd also like to see a more explicit comparison between the methods. For example, if we understand correctly, both the binomial test and the Bayes factor method can be applied to individuals that are heterozygous for J6. Wouldn't it make sense to directly compare the results between the methods for these two techniques? It seems that Figure 5a does some comparison (though note our questions about it below), but it's aggregated into a count that doesn't directly compare the per-subject results. There are also per-subject comparisons (e.g. Figure S3) but this doesn't give us the overall picture. Even if the results don't match perfectly, it'd be interesting to know more. E.g. do the results with huge Bayes factors tend to agree with the binomial-test results more than those with intermediate Bayes factors?
 - We are sorry for this confusion. The binomial deletion inference and the Bayesian haplotype inference methods are not comparable. The binomial method is applied before haplotype inference, and is able to detect genes that are deleted from both chromosomes. Once the deleted genes are detected, the next step is haplotype inference of the non-deleted genes. This is done by applying the Bayesian approach with an anchor gene (J6, D2-21, D2-8 or Vpooled). The method's certainty level is estimated by a Bayes factor. By sequentially combining both methods, a more accurate and full picture of the gene usage, allelic usage, and haplotype can be acquired. To avoid confusion we added text to the introduction that clarifies the relationship between the two methods.
- It's also worth thinking about what the overall biological results are from the analysis, one of which is a “mosaic structure.” However, the meaning of the word “mosaic” in this context wasn't immediately obvious to these computational biologists. Based on the p.3 text, we *think* that mosaic here refers to having one allele or another, but not both. If you want to have it in the title, please define in the abstract and introduction, and then provide additional

discussion about what might lead to these patterns (Are they just persisting haplotype blocks? Is there some reason why there should be exclusion? Can you describe why this is a surprising/interesting result?) From our perspective, though, the coolest part of this paper is that you have done the first large-scale haplotype inference, and an alternate title could focus on that.

- The title was changed to include the haplotype aspect of our work. The term mosaic is used to describe the pattern of tiled nearby gene deletions across the population. To make it clearer, we added text to the abstract explaining it. The exclusion between gene pairs V1-8 + V3-9, and V3-64D + V5-10-1, has been suspected to be a persisting haplotype block. Here we validate this suspicion in a large scale dataset, using a statistical framework that is able to handle haplotype analysis. We edited the text to reflect this.

- The organization of the paper methods is somewhat unusual, with a very short methods section, and a lot of methods appearing in the Results section or in figure legends. We're happy to accept this as a stylistic thing, but there are examples such as Figure S2 where methods important to the paper development are defined in a supplementary figure legend (!). This is not optimal for readability, and having to unpack motivation, methods, and results from this supplementary figure legend took a lot of effort. Having such material in the methods will allow for a more readable flow, starting with the goal of the analysis and then following with what was done and why.

- We thank the reviewer for pointing this out. We added the methodological details from the old legend of figure S2 as a section in methods, and removed them from the new S2 legend (now it is named figure S4).

- I (Erick) would love to see the inference part of this paper pushed farther in future work. Right now (Figure 5) you are estimating a Bayes factor for the presence of each V allele on the chromosome for each V gene individually. It would be really nice to get full haplotypes, and the confidence that we have in each one of them. This was the goal in the Elhanati... Walczak paper described below, although they didn't come up with statistical support for their haplotypes. This would end up being a more principled approach to the Vpooled work in the last section. Also, it would be nice to use the fact that the probability of misannotation (epsilon in your setting) depends a lot on what the actual underlying sequences are: a few confident calls for two rather different alleles means a bunch more than a bunch of uncertain calls for similar alleles. I think a fully Bayesian approach to haplotype inference is possible (we outline some ideas in our recent "Bayesian optimist" review paper). Of course, I'm not asking for additional work here, but I hope you will pursue these directions. You could mention such avenues in your Discussion.

-Thanks for the suggestion. We added text to the discussion about it.

- There are lots of figures that present genotypes of lists of subjects. Subjects are inherently unordered, which raises the question of how to order individuals in these figures. The sorting of the subjects of course has a large impact on our perception of whether there's patterns in the haplotypes or not. It seems like sometimes the individuals are ordered (Figure 2) and sometimes just ordered by number (E.g. Fig 3, Fig 5). We feel like some consistency would be good here, and lean towards wanting things sorted to see similarity between genotypes. Here's a way to do this if you want: calculate a pairwise distance matrix between the per-individual results (you are already sensibly using Jaccard in Fig S2), and then do a 1-D multidimensional scaling (there are packages for this in R and Python). Take the induced ordering from this to sort your samples. Then hopefully

we can see groups of haplotypes. I note that this is the goal of

<https://bitbucket.org/dmarnetto/haplostrips>

(<https://besjournals.onlinelibrary.wiley.com/doi/abs/10.1111/2041-210X.12747>), although it doesn't look like that software would actually be convenient for you to use.

- We agree that it is better to be consistent in this matter. We took your advice, reordered and accordingly renamed the samples in all figures based on a hierarchical clustering analysis performed in R using the heatmap function. We updated the methods section.

- Do we know the ethnicities of the donors? Particularly for S65 (different J alleles to everyone else), it would be interesting to know.

- For most donors, unfortunately, the ethnicities are not known. Since sample collection was done in Oslo, it is likely that the vast majority of individuals are Norwegians. In particular, S65 is known to be Norwegian (now S75 based on the new samples order).

- I (Erick) wasn't immediately able to find sequence data at the EGA by searching for "ERS2445766". Did I miss something, or will it be released upon submission of the paper? Also, is there a code repository associated with the paper?

- The sequence data will be released upon publication of the paper.

Specific comments:

- It is so much easier for us and you if you add line numbers to submitted manuscripts.

- Added.

Abstract:

- suggest "inference to IGHD- and IGHV-based," -> "inference to IGHD- and IGHV-based analysis," or some such.

- Edited.

- "We tested our method on the largest..." doesn't actually make it clear that you also generated this data set, which is pretty great!

- Thanks, edited to: "We generated the largest data set, to date, of naive B-cell repertoires, and tested our method on it".

p.1

- each B and T cell is unique? Hmm... debatable.

- Edited.

p. 2

- suggestion: "Only very few complete or partial"

- Edited.

- suggestion: “So far, a statistical framework”

- Edited.

- suggestion” “This enables looking also at the J distribution”

- Edited.

- suggestion: “We present evidence for”

- Edited.

Relative gene usage[...]

- it wasn't obvious to me staring at table S1 that including multiple assignments introduces biases. Perhaps you can clarify in the table legend or text?

- Text was added to the figure legend to clarify this point.

- It would be nice for sequential readers to introduce IK in a sentence or a phrase after you introduce your Bayesian method.

- Text was added to the legend of Figure 1.

- sentence starting “With the genotype step ~2% of the sequences” as written suggests that 98% of sequences assigned to genes removed during genotyping were not affected? I'm assuming it's a typo, which could be fixed by just removing one stray “that”, but suggest something more like: “~2% of sequences were initially assigned to genes that were removed during genotyping. They were thus reassigned to genes present in the subject.”

- Changed.

- suggestion: “in many individuals and involving multiple genes”

- Edited.

p. 3 Fig 1

- We like this figure, but feel that it needs to decide if it's an overview figure, or an example results figure. If it's an overview, then 1D should be simplified to make it easily understandable. Reducing the number of germline genes shown would help a lot, which would allow the reader to see cases of heterozygosity more clearly, and give room to add J6*02 and J6*03 labels on the left hand panel of 1D. If you want 1D to show results, then it should be split off to another figure giving a complete example result. Also, it wasn't obvious at first that the colors of the left-hand panel tell the complete story of the middle two panels (J6*02 and J6*03). If this is going to be an overview figure, then you could have arrows taking us from the count data over to the next two panels to clarify this point. Alternatively, you could drop those next two panels, add the J6*02 and J6*03 labels under the left and right halves of the left panel's x-axis, and add a little symbol showing which low-usage genes were called as deletions (e.g. a grey 'x' or something over the correspond

count bar). It was difficult to distinguish between the colors-- I'm certain I wouldn't be able to tell the difference between 03 next to a dark bar and 18 next to a light bar. Perhaps try <http://colorbrewer2.org/#type=qualitative&scheme=Set1&n=9> which has 8 colors and a gray?

- The figure is indeed intended to be illustrative. We adapted your suggestions.

- Some of the genes on the y-axis have single bars in their row, while others have two bars -- these must be heterozygous? It's confusing to discover this and have to guess, so it'd be nice if it was in the caption. Alternatively, why not just have two rows for genes that're heterozygous?

- Added: "The width of the count bar is inversely proportional to the number of alleles found on the chromosome".

p. 3

- Sentences after "Specifically:" -- suggest using i,ii,iii,iv rather than A,B,C,D, which also refer to figure panels and I don't think that the points and the panels correspond.

- Edited.

- V1-45, V4-28, D6-25: why is it not equally plausible that these occur only in some individuals and at very low prevalence? Also, it was confusing that in the text these genes were labeled as "non-functional" while in the figure legend they are indicated as being genes with more than 90% deletion. Isn't this an important distinction?

- We agree with this comment and added text to the results to clarify this issue.

- The take-home message about neighboring genes being deleted together as described in A-D could be more clearly introduced, perhaps with a topic sentence and then a clarification that A-D are examples of this pattern.

- We added a topic sentence and the suggested clarification.

- "Further exploration of possible clinical implications" and the previous "Here" sentence sure sound like a Discussion topic rather than a Results one.

- We removed the "Further" sentence and edited the "Here" sentence.

p. 5

Ig heavy chain gene heterozygosity landscape

- On the subject of genes with four alleles (i.e. that are presumably misnamed in IMGT) -- many readers would probably find it informative if there was some discussion of how IMGT goes about naming genes and alleles. Especially since such misnaming will have a distinct effect on all (?) of the results in this paper. In other words, all of the results talking about hetero vs homozygosity depend on the extent to which the IMGT distinction between genes and alleles is accurate. The acknowledgment in this paragraph (and at the top of p. 9) that IMGT names are not perfect is important, but it would be good to also discuss how frequent these miscategorizations are, and why you're confident that they're not common enough to materially affect your results.

#Andrew, can you please try to address this point?

- To establish a hot region for germline recombination, don't you need to look at more distant correlation between gene use or do some further statistical analysis? It seems that having a region with increased heterozygosity could occur even with a more even mix of ancient haplotypes.

- You are right, and we toned down the statement about the hot region for germline recombination.

- "Within this region, the three genes, V3-66, V3-64, and V4-61 appear as mostly homozygous. This is not the case, however, because there are many single chromosome deletions in these genes as shown in the following sections." It's not obvious to us that this use of "homozygous" lines up with your definition "We considered genes for which more than one allele is carried by an individual as heterozygous." That is, if a gene is present on one haplotype and deleted on another, isn't it still homozygous by your definition of heterozygous? In any case, more clarity would be appreciated here.

- We modified the text for clarity regarding this issue.

Fig 3

- "fraction of heterozygous genes out of all individuals"-- we feel that it would be clearer to explicitly describe the ratio in terms of a numerator and denominator.

- Edited.

p. 6

- Isn't this just the binomial test? I don't know what a binomial sign test is. The sign test is for paired data. There is a section in the Methods that describes "a special case of the binomial test" but what exactly is this?

- This is a binomial single sample sign test. See: Sprent, P. (1989), Applied Nonparametric Statistical Methods (Second ed.), Chapman & Hall, ISBN 0-412-44980-3.

We modified the terminology throughout the text to Binomial single sample sign test to make it clearer.

- Benjamini-Hochberg: the p-values after applying this procedure should be called q-values. p-values are defined by the fact that they are uniformly distributed under the null. q-values do not satisfy this property, but you are correct that they are the right tool for this job. It might be worth mentioning their interpretation in terms of false discovery rate.

- All adjusted p valued have been renamed to q values.

Fig 4

- We suggest dropping B, because A shows the same information in a much more information-rich way. If you want to keep the **, etc, you could just add it after the gene name. In fact, I'm not even sure that the box plots in A add to understanding-- there are an appropriate number of points for the scatter to show everything.

- We removed the bars from panel B, and kept the numbers and the **.

p. 8

Fig 5

- We were left scratching our head about this figure. If we understand correctly, this presents the deletion results inferred by haplotype and inferred by binomial test. So then each bin can have the same subject twice? If the intent is to make a comparison between the two methods, it could be done more directly.

- The binomial test is done **before** the haplotype inference step. For each individual and each gene the binomial test is applied first, to test if this gene is deleted. Only for non-deleted genes haplotype is inferred. Panel A shows the results of the two steps combined. Due to this comment and a comment raised by the other reviewer, we added text to the manuscript to clarify this point.

- caption: [typo] caption says both upper and lower panels are for j6*02

- Edited.

p. 9

Relative gene usage may [...]

- While I definitely agree that it's worth checking to see if gene usage can be used to detect single-chromosome deletions, a lot of the detail in this section about particular cases could be removed and summarized by a statement to the effect of "it doesn't really work". A quick glance at Fig 4 shows that allele usage ratios vary by a lot more than a factor of two, confirming the difficulty with such a method. It seems strange to pick the examples where it does seem like there is a separation and put them in a separate figure (6C)-- given this "multiple testing" situation should we believe that the observed separation is meaningful?-- the "it doesn't really work" conclusion seems to be tacitly admitted at the end of p. 10, where you say that the binomial test only works for deletion on both chromosomes (and that j6 anchor has to be used for single-chromosome deletions). If that's the conclusion, then we suggest being more clear about that from the start of this section.

- Thank you for pointing this out. Indeed, in its original form this observation was less statistically solid. Following your comment, we expanded this section with additional analyses. We derived a list of gene specific thresholds to predict single chromosomal gene deletions from relative gene usage of individuals without inferred haplotypes. Each threshold determines a certain sensitivity and specificity for inference of single chromosomal gene deletions. We changed Figure 5, and added supplementary figure S3, Supplementary table S3, and a method section.

Haplotype can be inferred [...]

- The language in this first paragraph can be tightened up and made more focused-- the current narrative structure is hard to follow. We suggest starting with a topic sentence that describes your goal (finding useful D genes) and then proceeding to results about how well these genes work for haplotype inference. Other details can be moved to the methods section.

- A topic sentence describing our goal was added. Text was shortened and edited.

- “in a chromosomal linkage map between the alleles of these two genes.” We assume you mean D2-21 and D2-8, but please specify.

- Edited.

p. 11

- “Due to the potential allele mix of the V_{pooled} approach” what do you mean by allele mix? That different subjects will have different combinations of V alleles?

- We use allele mix, as a term to describe a situation where contradicting chromosomal assignments of alleles is done by different V genes. We added text to clarify this point.

p. 13

- First sentence: you mean “genetic factors”

- Edited.

- You are missing a citation of Elhanati, Y., Sethna, Z., Marcou, Q., Callan, C. G., Jr, Mora, T., & Walczak, A. M. (2015). Inferring processes underlying B-cell repertoire diversity. Philosophical Transactions of the Royal Society of London. Series B, Biological Sciences, 370(1676). <https://doi.org/10.1098/rstb.2014.0243> who infer haplotypes (you have to see the supplementary material of this paper for details).

- Citation added.

p. 14

Library prep:

- golly, it’s only on p.14 that we learn that half the cohort has celiacs. Isn’t that worth describing earlier? I hope that you will look at association in future work (you could mention that).

- Please look at the replies to the editor’s and the other reviewer’s comments. Two supplementary figures and a text describing them have been added to the manuscript. No statistically significant differences were observed at the level of gene usage, gene deletions, and haplotypes.

- Is “Regional Ethical Committee” a typo?

- It is a typo indeed. Fixed.

- binomial test: - “the lowest relative frequency of this gene among all non-candidate samples” probably clearer to say “non-deletion-candidate samples” or something, it’s hard to remember what they’re candidates for.

- Edited.

- doesn't this whole procedure depend entirely on the (as far as I can tell) arbitrary choice of candidate frequency threshold? I.e. do the results change if we choose something other than 0.001/0.005 for v/d?

- Yes, of course. The cutoff is tightly related to sequencing depth and thus would affect differently various datasets. We chose the cutoff to reflect the minimum number of sequences in a sample (2000): in the smallest sample, if less than two reads are assigned to a particular gene, we test them for deletion. For the D gene we applied a larger threshold as the reliability of the assignment is lower. These thresholds should be subject to further research for their dependency on experimental factors. We added test to the methods section to explain this.

Data preprocessing:

- not all programs have version numbers

- Edited.

- I'm guessing that you ran partis with the germline inference code turned off, which is now run by default. It'd be worth mentioning.

- Edited.

- It reads a little strangely that first Tigger was applied, then Bayesian Tigger? Ref 28 describes some additional improvements to the core Tigger algorithm, and I'm guessing that this new code was run from the start.

- There is no difference between TigGER and TIGGER adapted for Bayesian approach in detecting novel alleles. The difference between the two is only at the genotyping step. We edited the citations for clarity.

- Binomial test: - suggest "usage of different genes" -> "usage of some genes"

- Edited.

Haplotype inference

- "Each sequence represents a unique recombination event"-- this is an assumption, especially given that we're just looking at heavy chains (IIUC there is proliferation of B cells just after heavy chain rearrangement to get multiple chances to rearrange light chains).

- We agree with this comment and changed the sentence to say " For simplification, we also assume that each sequence represents a...".

- "by the J allele that is present on them" suggestion: "by the J allele they contain"

- Edited.

- “X is a two dimensional vector with the number of sequences that this V allele appeared with ...” suggest: “X is a two-dimensional vector with components given by this V allele’s number of occurrences with each of the two J alleles”

- Edited.

- how was a Bayes factor of 1000 chosen as the cutoff? It’s a bunch stronger than the Kass-Raftery “decisive” cutoff from their foundational paper.

- This is true. There are sampling and experimental factors that affect the independence assumption of the Bayesian framework we used. To account for them and be conservative in the output of our methods, we set much higher thresholds for the Bayes factors. Applying our method, users can set the thresholds depending on their experimental considerations.

Bayesian TIGGER paper:

We understand that we aren’t meant to review this paper, but thought we would just add a few minor comments, which hopefully will be helpful.

Thanks!

- the formula for the posterior distribution is missing a P

- Edited.

- The posterior conditions on the data, but here it’s written using \vee , which is an “or”. Don’t you want a vertical bar?

- Yes, thanks!

Reviewers' comments:

Reviewer #1 (Remarks to the Author):

The authors have addressed some of my original concerns, but only some.

The addition of new data to Supplementary Table S1 listing the number of naive B cells sorted from each donor is helpful but I had hoped this would at least partially explain why the numbers of sequences used for analysis were so few for many of the donors. In particular for donors S50, S51, S52, S53, S54, S55, S56 and S57, but also donors S13, S24, S29, S43, S44, S65, S67, S70 and S75. The new data show that the number of starting cells was approximately the same for all samples. Yet, the total number of unique or single assignment VDJ sequences for these 17 donors are below 10,000, and for another 10 or so donors below 20,000. This is problematic. Overall, this low sequence depth is a major concern as it may well influence the results.

In response to my question about this, the authors added a sentence to the Discussion: "Other major factors that have a strong influence on our approach are the number and type of cells sequenced, and sequencing depth". This is not a satisfactory action to me. The authors should properly acknowledge that low expressed genes are particularly prone to erroneous identification of deletions and discuss that this problem is significant if the starting library is small. For example, V1-69-2, V-70D, V3-64, V3-43, V3-43D and V4-30-2 are low expressed genes for which the authors inferred homozygous deletions in several donors.

Given that all conclusions in the paper are from a single set of analysis using overall quite small libraries, the authors should perform an independent experiment with larger library sizes from some of the donors for whom they inferred deletions of low expressed genes. Without such confirmatory experiments, many of the specific results the authors present are unsubstantiated.

This also relates to my comment on Figure 4: "the authors do not discuss the large range in allelic preference observed between different individuals. What is the basis for this? In several cases this analysis is based on very few sequences assigned to the given alleles". The response: "The range of allelic preferences observed between different individuals is large, most likely due to factors related to their heterogeneous genetic background." This reply does not address the issue of sequence depth and the problem with inferences for low expressed genes. This also affects the data presented in Figure 6, now revised in response to reviewer 2. Here I also wonder why the data colored in gray labelled as "unknown" is shown? This does not add any information.

In response to the question from both myself and Reviewer 2 regarding how the two methods compare, the authors reply: "The binomial method is applied before haplotype inference, and is able to detect genes that are deleted from both chromosomes. Once the deleted genes are detected, the next step is haplotype inference of the non-deleted genes". This raises the question how the results would compare if the haplotype analysis was performed without the prior binomial method to infer homozygous gene deletions?

I appreciate the efforts to confirm the inferred deletions of V3-9, V1-8 and V5-10-1 and V3-64D through targeted genomic sequencing of some donors. However, as the authors state: "Due to high homology between the variable heavy chain genes, gene specificity of primers is often compromised". Therefore, the gel pictures provided in the new Supplementary Figure S2 should be accompanied by results from cloning and sequencing the PCR product. This is a minimal requirement and standard practice for this type of validation work even if this specific deletion pattern was reported previously.

Finally and importantly, one of the major selling points of the paper as outlined in the Abstract is that haplotype inference can be extended to IGHD-based anchoring. Yet these data are shown only as supplementary material (Figure S5) and a comparison of J and D anchoring is shown only for one individual. Here it is clear that while there is a good correlation for single chromosome deletions, the certainty level is very low for D-anchoring (due to many more D genes compared to J genes), which leads to the question of if it should be used. In my opinion, the authors should comment on the certainty level to give readers a fair message.

Additional specific comments:

Figure 2. The order of donors in Figure 2 was readjusted in response to a comment by Reviewer 2, but the Figure legend does not appear to be correctly updated. It still states: "Order of rows was determined by sorting the gene deletions first by V2-70D, then by V3-43D, and finally by V4-30-4". The legends also still states: "D gene labels marked in red represent indistinguishable genes due to high sequence similarity, therefore alignment call is less reliable", but there are no D genes labelled in red in the resubmitted version of the manuscript. This needs revision.

Figure 4. A closer look at the result of Figure 4 also raises some additional questions. Lines 125-127: "Out of 42 allele pairs (23 genes) that were tested, significant differences were found in 17 allele pairs (14 genes, see figure 4). In 10 allele pairs, the preferred allele was significantly more expressed than its partner in all individuals". Counting the allele pairs and numbers of genes, I find that the text should read: Out of 42 allele pairs (28 genes) that were tested, significant differences were found in 17 allele pairs (13 genes, see figure 4). In 14 (or 15?) allele pairs, the preferred allele was significantly more expressed than its partner in all individuals". Please correct.

Figure S1. The legend has not been updated to fit the table. Also, the number of sequences for donor S54 is below 2000, which is below the stipulated criterion for exclusion.

Reviewer #2 (Remarks to the Author):

Since the authors have addressed almost every comment we made, we have very few comments on this edited draft, and thank the authors for their careful responses. A few minor points follow.

Line 18: "preformed" seemed like a surprising adjective for something dynamically generated

Line 61: suggest "is the first of its kind in size and accuracy"

Fig 1 caption: thanks for adding the heterozygosity clarification, but I think it should be that the height (or thickness, but not width) is inversely proportional to allele count?

Line 173: The new approach to get single chromosome deletions from gene usage seems totally sensible, but I had one comment. If I'm understanding correctly (which I may not be!) this amounts to using the present sample of individuals as a training set to figure out a threshold for each gene. Now I haven't thought too much about what these per-gene thresholds would depend on, but it seems quite possible that they would vary across different human populations. In particular, using a likely very homogenous sample of Scandinavians might not be very accurate when applied to a much more diverse sample of, say, African individuals. If this makes sense, you might want to in future check that

the thresholds don't change too much when data from more individuals becomes available.

Line 338: Thanks for including some ideas about future Bayesian development, though I feel like the point about estimating confidence for complete haplotypes rather than per-gene Bayes factors is a little more meaty. See my original comment for more details. Also, I (Erick) feel a little awkward that you named Olson and Matsen in the manuscript in response to a review comment. I wasn't asking for a citation. If it seems worthwhile to cite us, that's ok, but there's no need to include our names.

P. 5 [...]

#Andrew, can you please try to address this point

Looks like you missed answering this question about the prevalence of mislabeled gene/allele distinctions in IMGT? While we don't think that it's an important enough point to be worth delaying publication, if you have a chance to mention it we feel it would be enlightening to some readers. I should note that you do touch on this in the new paragraph in the "Relative gene usage may..." section (line 180) in a way that was quite interesting.

Reviewers' comments:

Reviewer #1 (Remarks to the Author):

The authors have addressed some of my original concerns, but only some.

We appreciate your comprehensive review of our manuscript. We considered seriously all of your comments, and made efforts to address them. We believe that the additional analyses provided here fully clarify these concerns.

The addition of new data to Supplementary Table S1 listing the number of naïve B cells sorted from each donor is helpful but I had hoped this would at least partially explain why the numbers of sequences used for analysis were so few for many of the donors. In particular for donors S50, S51, S52, S53, S54, S55, S56 and S57, but also donors S13, S24, S29, S43, S44, S65, S67, S70 and S75. The new data show that the number of starting cells was approximately the same for all samples. Yet, the total number of unique or single assignment VDJ sequences for these 17 donors are below 10,000, and for another 10 or so donors below 20,000. This is problematic. Overall, this low sequence depth is a major concern as it may well influence the results.

Following your comment, we looked again at the numbers. There are a few technical and biological factors that could potentially cause this kind of diversity. Technical steps like cDNA synthesis, PCR amplification, adapter ligation, library cleaning, and sequencing vary between libraries. In a similar naïve B cell dataset with 13 samples generated independently in another lab with a slightly different sequencing protocol utilizing UMIs (Vander Heiden, J.A., et al., Dysregulation of B Cell Repertoire Formation in Myasthenia Gravis Patients Revealed through Deep Sequencing, *The Journal of Immunology*, 1601415, 2017), the number of reads ranged between 1817-50933, despite the fact that, like in our experiment, the number of starting cells was approximately the same for all these samples. The range of number of sequences in their study aligns with ours.

Importantly, we constructed our libraries using UMI's, and used very stringent filtering criteria in order to prevent inclusion of artificial sequences. This reduces the final number of sequences per library but greatly increases the reliability of the dataset. Taking into account all these factors, we believe that our library sizes are within the typical size range obtained using state of the art technology.

To estimate the effect of sequencing depth, we performed the following two analyses:

1. Only samples with more than 20K sequences were taken into account. We re-generated figure 2 and figure 4 (figure R1 and R2 below, respectively), and obtained similar results with identical conclusions. E.g., i) In 29 of the 30 individuals that lack V2-70D, the adjacent gene V1-69-2, is also deleted. ii) In 10 of the 11 individuals that lack V4-30-2, the adjacent genes: V4-30-4 and V3-30-3 are also deleted. iii) Out of 39 individuals that lack V3-43D, 38 lack also V4-38-2.

2. Only samples with more than 30K sequences were taken into account. We generated new smaller samples by sub-sampling 2K/5K/10K sequences from each such sample, and recorded the number of deletions in a single or both chromosomes (figure R3 below). This process was repeated 10 times and the presented results are the averages of these 10

realizations. The number of inferred deletions is indeed affected by the sample depth. The smaller the sample, the fewer deletions are inferred. Notably, no new deletions are discovered in the smaller samples. Thus, the sequencing depth indeed has an impact on the results received, however the 10K and 5K libraries maintain the information inferred from the original size library. We therefore conclude that smaller size libraries provide reliable results, although they may not find all of the deletions.

In summary, our method was designed to confront the limitations of current sequencing technologies so that it can be used on typical depth datasets. The Bayes factor reflects the certainty level in cases of low numbers of sequences, and different thresholds can be imposed on it to obtain different levels of reliability.

We added text to the discussion about this issue.

Figure R1: Gene deletion inference by relative gene usage. Only samples with sequence depth above 20K are shown. Analog to figure 2 from the main text of the manuscript.

Figure R2: Gene deletion inference along each chromosome. Only samples with sequence depth above 20K are shown. Analog to figure 5 from the main text of the manuscript.

Figure R3: Sub-sampling different sequence depths. (A) Deletion inference from the binomial test. Each dot represents an individual. The X axis is the number of deletions inferred from the original data. The Y axis is the average number of deletions inferred in the 10 sub-sampling repetitions performed for each individual. Each panel shows a different sequence depth. (B) Single chromosome deletions inferred from haplotype inference. Each dot represents an individual. The color of the dot represents the IK threshold value used to infer the single chromosome deletion. The X axis is the number of deletions detected for the original sequence depth. The Y axis is the average number of single chromosome deletions inferred in the 10 sub-sampling repetitions performed for each individual. Each panel shows a different sequence depth. (C) The Jaccard distance between the haplotype inferred for the original sequence depth and the sub-sampled samples. Each dot represents the average distance inferred in the 10 repetitions of sub-sampling performed for each individual. The color of the dots represents the sequence depth sampled. The X axis is the fraction of the unknown assignments (the 'Unknown' label in figure 7). (D) The ratio between the mean gene usage in the sub-sampled data and the original samples. For each dot the standard error is shown.

In response to my question about this, the authors added a sentence to the Discussion: “Other major factors that have a strong influence on our approach are the number and type of cells sequenced, and sequencing depth”. This is not a satisfactory action to me. The authors should properly acknowledge that low expressed genes are particularly prone to erroneous identification of deletions and discuss that this problem is significant if the starting library is small. For example, V1-69-2, V-70D, V3-64, V3-43, V3-43D and V4-30-2 are low expressed genes for which the authors inferred homozygous deletions in several donors.

Thank you, this is indeed an important remark. Due to the high variance in gene usage, it is very hard to visualize the bimodal distribution, which is the basis for the homozygous deletion binomial test decision. To evaluate if the deletions in low expressed genes are the result of the cutoff set by us or a true bimodal distribution, we took a closer look at the six genes suggested above, as well as six additional genes with similar gene usage distribution patterns. For these genes we estimated two gamma distributions based on a 0.001 cutoff used in the binomial test for V genes deletion detection, which is now described in the manuscript. We added figure S10 to the supplementary, where we generated empirical cumulative distribution function curves of the gene usage, box plots of the gene usage with the inferred deletions, and histograms with the estimated gamma distributions. To estimate a deletion in a specific gene in a specific sample, we expect a clear visible separation between the two gamma distributions. From this newly added figure S10, in two of the genes (V3-43, V3-20) we cannot observe a clear bimodal distribution and therefore should treat those deletions with greater caution. However, for the rest of the genes, we observed two distinct distributions that indicate high reliability of the deletions inferred by the binomial test, even for low usage genes. Following this comment and the subsequent analysis, we marked in red the two genes in figure 2 in which the deletion inferences did not come from bimodal distributions, and added text to the methods.

Given that all conclusions in the paper are from a single set of analysis using overall quite small libraries, the authors should perform an independent experiment with larger library sizes from some of the donors for whom they inferred deletions of low expressed genes. Without such confirmatory experiments, many of the specific results the authors present are unsubstantiated.

We appreciate your comment. In addition to our previous responses, we took another action, which we believe resolves this issue.

We repeated our analysis with high coverage libraries from two independent public datasets:

- DS1: Vander Heiden, J.A., et al., Dysregulation of B Cell Repertoire Formation in Myasthenia Gravis Patients Revealed through Deep Sequencing, *The Journal of Immunology*, 1601415, 2017
- DS2: DeWitt WS, Lindau P, Snyder TM, Sherwood AM, Vignali M, Carlson CS, et al. (2016) A Public Database of Memory and Naive B-Cell Receptor Sequences. *PLoS ONE* 11(8): e0160853. <https://doi.org/10.1371/journal.pone.0160853>

The first dataset of Myasthenia Gravis (MG) patients, DS1, contains sorted naive B-cells with UMI's, pre-processed as described in the methods section of the paper. We took these data, and genotyped and haplotyped them. This resulted in 10 out of 13 samples with more than

2000 reads, which were suitable for further analysis. Utilizing the gene usage distribution of the 94 samples from our dataset as reference, we detected deletion patterns similar to the ones in observed in our manuscript (see figure R4 below): I) For 2 out of the 10 samples V4-30-4 and V3-30-3 were deleted. II) Out of 3 individuals that lacked V3-43D, 2 lacked also V4-38-2. III) Two pairs of genes, V3-9 and V1-8, and V5-10-1 and V3-64D, were deleted in a mutually exclusive manner in all samples. D gene deletions were not observed in the MG data, and therefore are not shown. Four out of the ten samples were heterozygous for J6 and their single chromosomal deletion also resembles the pattern observed in our manuscript (figure R5 below).

Figure R4: DS1 dataset gene deletion inference by relative gene usage, analog to figure 2 from the main text of the manuscript.

Figure R5: J6 heterozygous samples from the DS1 dataset. Gene deletion inference along each chromosome, analog to figure 5 from the main text of the manuscript.

DS2 is the deepest dataset of naive B-cells to date. Antibodies from three individuals were sequenced with millions of reads each. However, these reads were produced using a library preparation protocol that covers only a short region of the IGHV genes. Therefore, this data set is heavily prone to gene and allele mis-assignments. In addition, no UMI's were used, which greatly reduces the libraries' reliability. To overcome these limitations, we used two different aligners (IgBLAST and IMGT/HighV-QUEST) and tested the agreement between their gene assignments. The agreement between the aligners for all three individuals was ~62%, compared with ~97% for the full length IGHV sequences used in our manuscript. Because of the low level of agreement between the aligners for the DS2 data, we estimated the alignment reliability for each IGHV gene independently. We calculated a reliability score for each gene by averaging over the agreements between the aligners for all individual sequences mapped to this gene. An individual agreement score between the aligners for a given sequence was:

- 0 if there was no or partial agreement between IgBLAST and IMGT,
- 1 if there was a single assignment with full agreement,
- 1 over the number of unique genes mapped to this sequence, if there were several assignments with full agreement.

So for example if we analyzed sequences X, Y and Z: X was mapped only to 1-69 in both aligners, Y was mapped to 1-69 and 1-8 in IMGT and to 1-69 in IgBLAST, and Z was mapped to 1-69, 3-9 and 1-8 in both aligners, then the cumulative reliability score for 1-69 would be 1 (from X) + 0 (from Y) + $1/3$ (from Z) / $(1+1/2+1/3)$ (sum of 1 over the number of genes

aligned to each sequence) = 8/11. The reliability score for each gene is shown in the following graph:

Figure R6: Reliability score for each gene, according to the agreement between IgBLAST and IMGT in gene assignment. The red dashed line represent the threshold (0.05) below which genes were filtered out.

For deletion inference, we used one individual (D3 in the original annotations of the paper), who was heterozygous for IGJ6. We genotyped the IGHD and IGJ genes, and discarded all non-functional mutated sequences. In addition, only sequences with a single gene assignment for both IGHD and IGHV were used. This filtering left us with 1.6M sequences.

We sub-sampled this extremely high coverage data, to generate smaller samples for which we inferred deletions. A similar deletion pattern was observed for all sub-sampling (figure R7). For the small sub-samples below 10K sequences some of the deletions were not detected. However, no additional deletions to the ones discovered in the large datasets were observed in the smaller datasets. Therefore we conclude that although small samples might result in undiscovered deletions, the deletions that are discovered in the small samples have a high reliability. We added text to the discussion to clarify this point.

For the inference of single chromosome deletions, we repeated the sub-sampling procedure, and came to an analogous conclusion (figure R8).

Figure R7: Binomial test for gene deletion detection applied to decreasing library sizes. Each column corresponds to an IGHV gene, and each row to a sample size. For each sample size 100 realizations were generated, and the intensity of the color (blue) reflects the number of realizations in which a deletion was inferred (see color key). In black are genes that were eliminated from the analysis due to discrepancies between IgBLAST and IMGT assignments (see section above).

Figure R8: Single gene deletion inference by IGHJ6 haplotype in decreasing library sizes, analog to figure 5 from the main text of the manuscript. Here, each row corresponds to a single realization of a particular sample size. In black are genes that were eliminated from the analysis due to discrepancies between IgBLAST and IMGT assignments (see section above).

These two analyses demonstrate the robustness of our method in terms of single and double chromosome deletion detection, and that the results are not significantly affected by low to moderate sample sizes.

This also relates to my comment on Figure 4: “the authors do not discuss the large range in allelic preference observed between different individuals. What is the basis for this? In several cases this analysis is based on very few sequences assigned to the given alleles”. The response: “The range of allelic preferences observed between different individuals is large, most likely due to factors related to their heterogeneous genetic background.” This reply does not address the issue of sequence depth and the problem with inferences for low expressed genes.

Following this comment and the comments before about low sequencing depth and low gene usage, we included in the current version of the analysis only samples with more than 10K sequences and genes with more than 1% usage. The results did not change dramatically. The variation in the bias between individuals is still present and, as indicated in the discussion, might be due to individual variations in the RSS and epigenetic modifications.

This also affects the data presented in Figure 6, now revised in response to reviewer 2. Here I also wonder why the data colored in gray labelled as “unknown” is shown? This does not add any information.

The “unknown” shown in figure 6E refers to samples which are not heterozygous for J6, and therefore single chromosome deletions could not be inferred.

In response to the question from both myself and Reviewer 2 regarding how the two methods compare, the authors reply: “The binomial method is applied before haplotype inference, and is able to detect genes that are deleted from both chromosomes. Once the deleted genes are detected, the next step is haplotype inference of the non-deleted genes”. This raises the question how the results would compare if the haplotype analysis was performed without the prior binomial method to infer homozygous gene deletions?

The binomial test enables us to filter out genes that are inferred as deleted, and prevent them from entering the haplotype analysis. These deleted genes appear with very low frequencies. If these genes enter the haplotype analysis, they will have extremely low certainty levels that will most likely result in defining them as “unknown”. For example, in figure 7A we show the J6 haplotype results. V4-30-2 did not pass the binomial test in this specific sample (not deleted), and the haplotype results could not be determined for this gene on both chromosomes (light gray, “unknown”). V2-70 did not pass the binomial test in this specific sample as well. Since the number of reads associated with this gene is very low but not zero (3 in this case), the haplotype on one chromosome was inferred but with an extremely low certainty level. If genes that were determined as deleted by the binomial test were included in the haplotype analysis, they would show similar haplotype results. For example V4-38-2 will resemble the haplotype of V4-30-2, and V4-28 will resemble the haplotype of V2-70.

I appreciate the efforts to confirm the inferred deletions of V3-9, V1-8 and V5-10-1 and V3-

64D through targeted genomic sequencing of some donors. However, as the authors state: "Due to high homology between the variable heavy chain genes, gene specificity of primers is often compromised". Therefore, the gel pictures provided in the new Supplementary Figure S2 should be accompanied by results from cloning and sequencing the PCR product. This is a minimal requirement and standard practice for this type of validation work even if this specific deletion pattern was reported previously.

We repeated these experiments, and obtained convincing evidence that the deletion candidates are indeed deleted. We added the sequencing product table to the manuscript.

Finally and importantly, one of the major selling points of the paper as outlined in the Abstract is that haplotype inference can be extended to IGHD-based anchoring. Yet these data are shown only as supplementary material (Figure S5) and a comparison of J and D anchoring is shown only for one individual. Here it is clear that while there is a good correlation for single chromosome deletions, the certainty level is very low for D-anchoring (due to many more D genes compared to J genes), which leads to the question of if it should be used. In my opinion, the authors should comment on the certainty level to give readers a fair message.

This is true. Following your comment we added a new figure to the manuscript (figure 7), showing the haplotype inference both from J6 and D2-21. We compared both inferences using clustering (as currently described in the methods) and showed the similarity between them. In addition, we revised figure S5 to show the comparison between J6 and D2-8, using the same clustering method.

Additional specific comments:

Figure 2. The order of donors in Figure 2 was readjusted in response to a comment by Reviewer 2, but the Figure legend does not appear to be correctly updated. It still states: "Order of rows was determined by sorting the gene deletions first by V2-70D, then by V3-43D, and finally by V4-30-4". The legends also still states: "D gene labels marked in red represent indistinguishable genes due to high sequence similarity, therefore alignment call is less reliable", but there are no D genes labeled in red in the resubmitted version of the manuscript. This needs revision.

Text corrected. D genes now labeled by light gray (corrected in text). Added to the legend that deletions cannot be inferred for two low usage genes (see comments above), and marked those in red. The methods was updated accordingly (binomial test section).

Figure 4. A closer look at the result of Figure 4 also raises some additional questions. Lines 125-127: "Out of 42 allele pairs (23 genes) that were tested, significant differences were found in 17 allele pairs (14 genes, see figure 4). In 10 allele pairs, the preferred allele was significantly more expressed than its partner in all individuals". Counting the allele pairs and numbers of genes, I find that the text should read: Out of 42 allele pairs (28 genes) that were tested, significant differences were found in 17 allele pairs (13 genes, see figure 4). In 14 (or 15?) allele pairs, the preferred allele was significantly more expressed than its partner in all individuals". Please correct.

Corrected

Figure S1. The legend has not been updated to fit the table. Also, the number of sequences for donor S54 is below 2000, which is below the stipulated criterion for exclusion.

Thanks for pointing this out. We indeed mistakenly took out two wrong samples from the analysis. We now replaced these small samples by another sample (>2K) and therefore the samples were renamed in this version based on the new deletion clustering analysis.

Reviewer #2 (Remarks to the Author):

Since the authors have addressed almost every comment we made, we have very few comments on this edited draft, and thank the authors for their careful responses.

Thank you for all the constructive comments you made that helped us to further improve the manuscript.

A few minor points follow.

Line 18: "preformed" seemed like a surprising adjective for something dynamically generated

We deleted the word "preformed".

Line 61: suggest "is the first of its kind in size and accuracy"

Thanks. We are happy to integrate your suggestion in the text.

Fig 1 caption: thanks for adding the heterozygosity clarification, but I think it should be that the height (or thickness, but not width) is inversely proportional to allele count?

We modified the text to say "thickness" instead of "width".

Line 173: The new approach to get single chromosome deletions from gene usage seems totally sensible, but I had one comment. If I'm understanding correctly (which I may not be!) this amounts to using the present sample of individuals as a training set to figure out a threshold for each gene. Now I haven't thought too much about what these per-gene thresholds would depend on, but it seems quite possible that they would vary across different human populations. In particular, using a likely very homogenous sample of Scandinavians might not be very accurate when applied to a much more diverse sample of, say, African individuals. If this makes sense, you might want to in future check that the thresholds don't change too much when data from more individuals becomes available.

You are right in pointing this out and we added a sentence to the discussion to reflect it.

Line 338: Thanks for including some ideas about future Bayesian development, though I feel like the point about estimating confidence for complete haplotypes rather than per-gene

Bayes factors is a little more meaty. See my original comment for more details. Also, I (Erick) feel a little awkward that you named Olson and Matsen in the manuscript in response to a review comment. I wasn't asking for a citation. If it seems worthwhile to cite us, that's ok, but there's no need to include our names.

OK. Your names were removed from the text but we kept the citation as it is very relevant.

P. 5 [...]

#Andrew, can you please try to address this point

Looks like you missed answering this question about the prevalence of mislabeled gene/allele distinctions in IMGT? While we don't think that it's an important enough point to be worth delaying publication, if you have a chance to mention it we feel it would be enlightening to some readers. I should note that you do touch on this in the new paragraph in the "Relative gene usage may..." section (line 180) in a way that was quite interesting.

You are right and we are sorry for skipping this point in our 'response to reviewers'. As you noted correctly, we did address some of the issues raised by modifications to the manuscript. The reasoning behind these changes should have been explained, particularly since not all the comments were addressed. In our modifications to the manuscript, we have attempted to give some indication of the magnitude of the problem, but it is difficult to be more specific than we have been. We do not believe that the problem compromises our analysis. Problems are revealed by anomalies in the data, and as discussed, these are seen in our data in the relative usage of V4-61 in individuals with a single chromosomal deletion. Other sets of highly similar gene sets - such as IGHV3-30, IGHV3-33, IGHV3-30-3 and IGHV3-30-5- do not reveal anomalies in our data. Anomalies are also absent in our data in recognized gene duplications such as IGHV3-23 and IGHV3-23D.

By highlighting the V4-4/4-59/4-61 confusion, we hope that we have drawn attention to the possibility of rare problems with other gene sets, and that this will serve as a warning to researchers as they continue to explore variation within different human populations.

Unfortunately we are unable to discuss the IMGT nomenclature, as requested. The IMGT system includes thousands of pages of on-line documentation, but critical documentation is missing. The principles behind the IMGT nomenclature are provided, but there is no information available that documents the evidence that led to the naming of specific sequences. Critical to an understanding of the IGHV4-4/4-59/4-61 confusion is the history of how IGHV4-59*08 came to be named. There is no available documentation of this, and we have not been able to clarify this by direct communication with the IMGT leadership.

Reviewers' comments:

Reviewer #1 (Remarks to the Author):

The authors have performed a number of additional tests in this revision and I appreciate their careful response. I will restrict the current review to a few issues, which I believe remain to be addressed. They are:

On page 7 in the revised manuscript, the authors describe the results from Sanger sequencing of VH3-64D in subjects S42 and S49, performed in response to one of my previous comments. They find an allelic variant of the IMGT allele VH3-64D*06 with a mutation at position g258>t (Table S3). Overall, the potential presence of novel alleles (=non-IMGT alleles) is important for the conclusions drawn in this study and such alleles could be relatively common as the authors now find. The Methods section (Data preprocessing and genotyping) mentions that 25 novel alleles were found (+ the one mentioned above?), but it is unclear if these novel alleles were taken into consideration in all assignments?

The assignments in most figures show genes rather than alleles, which makes things fuzzy. If the assignments in Figure 3 do not include novel alleles, a gene may appear homozygous (or deleted depending on the level of divergence of the novel allele from closest known IMGT allele) whereas in fact the subject is heterozygous for the gene. This distinction should be described in more detail. If novel alleles are not taken into account and sequences < 3 nt different from known IMGT alleles are assigned to the closest IMGT allele, the level of homozygosity for this gene in this subject will be overestimated. On page 2 (line 72), it is stated that sequences with > 3 mutations in their V region were excluded from the analysis. Does this only concern sequences that are > 3 mutations different from known IMGT alleles present in the subject or also sequences that are > 3 mutations different from novel alleles present in the given subject?

Figure 3 is not consistent with Figure 5 since it concludes erroneously that V genes are present in either a homozygous or a heterozygous form. In Figure 5, the authors use J-anchored haplotype analysis to infer single chromosome deletions of a number of V genes. Thus, identification of just one allele of a given gene in the NGS data does not automatically mean homozygosity as is currently concluded in Figure 3. This leads to an over-estimation of the level of homozygosity. I understand that Figure 3 comes before Figure 5, but I strongly recommend that Figure 3 is revised since it adds confusion in its current form. Please also clarify what N/A represents in Figure 3.

On page 16, line 339, it is stated that V3-43D and V4-38-2 are not adjacent to each other, while according to IMGT they are. What is correct?

Please add total numbers of paired sequencing reads for the different libraries in Table S1.

Reviewer #2 (Remarks to the Author):

This reviewer has no further comment for this manuscript.

Reviewers' comments:

Reviewer #1 (Remarks to the Author):

The authors have performed a number of additional tests in this revision and I appreciate their careful response.

Thank you for all your comments and thorough review, which greatly contributed to the manuscript.

I will restrict the current review to a few issues, which I believe remain to be addressed. They are:

On page 7 in the revised manuscript, the authors describe the results from Sanger sequencing of VH3-64D in subjects S42 and S49, performed in response to one of my previous comments. They find an allelic variant of the IMGT allele VH3-64D*06 with a mutation at position g258>t (Table S3). Overall, the potential presence of novel alleles (=non-IMGT alleles) is important for the conclusions drawn in this study and such alleles could be relatively common as the authors now find. The Methods section (Data preprocessing and genotyping) mentions that 25 novel alleles were found (+ the one mentioned above?), but it is unclear if these novel alleles were taken into consideration in all assignments?

We agree that allelic variants that are not included in the IMGT database are important for all AIRR seq analyses, and in particular for genotype and haplotype inference. Throughout the manuscript we have considered these new variants in all analysis steps, and indeed 25 novel alleles were discovered including the allele discovered by Sanger sequencing as stated in the methods section.

The assignments in most figures show genes rather than alleles, which makes things fuzzy. If the assignments in Figure 3 do not include novel alleles, a gene may appear homozygous (or deleted depending on the level of divergence of the novel allele from closest known IMGT allele) whereas in fact the subject is heterozygous for the gene.

This is correct. As mentioned above, novel alleles were taken into account and appear in the genotype throughout the manuscript, including Figure 3.

This distinction should be described in more detail. If novel alleles are not taken into account and sequences < 3 nt different from known IMGT alleles are assigned to the closest IMGT allele, the level of homozygosity for this gene in this subject will be overestimated. On page 2 (line 72), it is stated that sequences with > 3 mutations in their V region were excluded from the analysis. Does this only concern sequences that are > 3 mutations different from known IMGT alleles present in the subject or also sequences that are > 3 mutations different from novel alleles present in the given subject?

Mutations were counted after re-alignment against a personal germline reference, which was constructed following genotyping, and designed to include novel alleles. This means that if a novel allele has more than three mutations from a known germline and is part of the genotype, the personal reference includes the novel sequences. Following, re-alignment, if a certain sequence exactly matches a novel allele, it is considered to have zero mutations.

Figure 3 is not consistent with Figure 5 since it concludes erroneously that V genes are present in either a homozygous or a heterozygous form. In Figure 5, the authors use J-anchored haplotype analysis to infer single chromosome deletions of a number of V genes. Thus, identification of just one allele of a given gene in the NGS data does not automatically mean homozygosity as is currently concluded in Figure 3. This leads to an over-estimation of the level of homozygosity. I understand that Figure 3 comes before Figure 5, but I strongly recommend that Figure 3 is revised since it adds confusion in its current form.

You are correct, and we thank you for this comment. Cases in which single chromosomal deletions are detected by haplotyping can be counted as deletion polymorphisms. The order of figures makes it challenging to add this information into figure 3. However, since we agree it adds important information, we added a supplementary figure (S2), that shows these deletion polymorphisms alongside other types of heterozygosity.

Please also clarify what N/A represents in Figure 3.

This has now been clarified in the figure legends.

On page 16, line 339, it is stated that V3-43D and V4-38-2 are not adjacent to each other, while according to IMGT they are. What is correct?

According to the table from IMGT (<http://www.imgt.org/IMGTrepertoire/index.php?section=LocusGenes&repertoire=GeneOrder&species=human&group=IGH>) there are 2 pseudo genes between V3-43D and V4-38-2. We added this clarification to the text.

Please add total numbers of paired sequencing reads for the different libraries in Table S1.

These numbers were added to the table.

Reviewer #2 (Remarks to the Author):

This reviewer has no further comment for this manuscript.

REVIEWERS' COMMENTS:

Reviewer #1 (Remarks to the Author):

The authors have added a new Supplementary Figure 2, which is helpful.

I am still unclear about whether the 25 new VH alleles detected by TigGER (described on page 17) are available to readers through EGA or elsewhere. This should be clearly stated in the text.

Reviewer #1:

As to the request of the reviewer #1, we confirm that we will deposit the newly inferred allele sequences in a public database, but will do so after necessary DNA sequencing verification. The scope of the current paper is characterization of structural variations in the antibody heavy chain gene locus across the population, and we consider reporting sequences of all the newly inferred alleles to be beyond the scope of the paper. The identification and sequence verification of novel alleles of immunoglobulin heavy and light chain genes is ongoing work in Yaari/Sollid labs, and in this effort we are also employing computational tools parallel to TigGER, i.e. IgDiscover and Partis, in the search of novel alleles. This task is by no means trivial. It will require much time and work, and it will be a paper on its own merit. If desired, we can add this or a similar text to the paper: "These newly inferred alleles are in the process of validation, and will be published elsewhere."